# Wolfpack Adversarial Attack for Robust Multi-Agent Reinforcement Learning

**Sunwoo Lee** [1]  **Jaebak Hwang** [1]  **Yonghyeon Jo** [1]  **Seungyul Han** [1*]

## Abstract

Traditional robust methods in multi-agent reinforcement learning (MARL) often struggle against coordinated adversarial attacks in cooperative scenarios. To address this limitation, we propose the Wolfpack Adversarial Attack framework, inspired by wolf hunting strategies, which targets an initial agent and its assisting agents to disrupt cooperation. Additionally, we introduce the Wolfpack-Adversarial Learning for MARL (WALL) framework, which trains robust MARL policies to defend against the proposed Wolfpack attack by fostering systemwide collaboration. Experimental results underscore the devastating impact of the Wolfpack attack and the significant robustness improvements achieved by WALL. Our code is available at https://github.com/sunwoolee0504/WALL.

## 1. Introduction

Multi-agent Reinforcement Learning (MARL) has gained attention for solving complex problems requiring agent cooperation (Oroojlooy & Hajinezhad, 2023) and competition, such as drone control (Yun et al., 2022), autonomous navigation (Chen et al., 2023), robotics (Orr & Dutta, 2023), and energy management (Jendoubi & Bouffard, 2023). To handle partially observable environments, the Centralized Training and Decentralized Execution (CTDE) framework (Oliehoek et al., 2008) trains a global value function centrally while agents execute policies based on local observations. Notable credit-assignment methods in CTDE include Value Decomposition Networks (VDN) (Sunehag et al., 2017), QMIX (Rashid et al., 2020), which satisfies the Individual-Global-Max (IGM) condition ensuring that optimal joint actions align with positive gradients in global and individual value functions, and QPLEX (Wang et al., 2020b), which encodes IGM into its architecture. However,

CTDE methods face challenges from exploration inefficiencies (Mahajan et al., 2019; Jo et al., 2024) and mismatches between training and deployment environments, leading to unexpected agent behaviors and degraded performance (Moos et al., 2022; Guo et al., 2022). Thus, enhancing the robustness of CTDE remains a critical research focus.

To improve learning robustness, single-agent RL methods have explored strategies based on game theory (Yu et al., 2021), such as max-min approaches and adversarial learning (Goodfellow et al., 2014; Huang et al., 2017; Pattanaik et al., 2017; Pinto et al., 2017). In multi-agent systems, simultaneous agent interactions introduce additional uncertainties (Zhang et al., 2021b). To address this, methods like perturbing local observations (Lin et al., 2020), training with adversarial policies for Nash equilibrium (Li et al., 2023a), adversarial value decomposition (Phan et al., 2021), and attacking inter-agent communication (Xue et al., 2021) have been proposed. However, these approaches often target a single agent per attack, overlooking interdependencies in cooperative MARL, making them vulnerable to scenarios where multiple agents are attacked simultaneously.

To overcome the vulnerabilities posed by coordinated adversarial attacks in MARL, we propose the Wolfpack adversarial attack framework, inspired by wolf hunting strategies. This approach disrupts inter-agent cooperation by targeting a single agent and subsequently attacking the group of agents assisting the initially targeted agent, resulting in more devastating impacts. Experimental results reveal that traditional robust MARL methods are highly susceptible to such coordinated attacks, underscoring the need for new defense mechanisms. In response, we also introduce the Wolfpack-Adversarial Learning for MARL (WALL) framework, a robust policy training approach specifically designed to counter the Wolfpack Adversarial Attack. By fostering system-wide collaboration and avoiding reliance on specific agent subsets, WALL enables agents to defend effectively against coordinated attacks. Experimental evaluations demonstrate that WALL significantly improves robustness compared to existing methods while maintaining high performance under a wide range of adversarial attack scenarios. The key contributions of this paper in constructing the Wolfpack Adversarial Attack are summarized as follows:

- A novel MARL attack strategy, **Wolfpack Adversarial Attack**, is introduced, targeting multiple agents simul-

---

[1]Graduate School of Artificial Intelligence, UNIST, Ulsan, South Korea. Correspondence to: Seungyul Han <syhan@unist.ac.kr>.

*Proceedings of the 42$^{nd}$ International Conference on Machine Learning*, Vancouver, Canada. PMLR 267, 2025. Copyright 2025 by the author(s).

taneously to foster stronger and more resilient agent cooperation during policy training.

- **The follow-up agent group selection** method is proposed to target agents with significant behavioral adjustments to an initial attack, enabling subsequent sequential attacks and amplifying their overall impact.

- **A planner-based attacking step selector** predicts future $Q$-value reductions caused by the attack, enabling the selection of critical time steps to maximize impact and improve learning robustness.

## 2. Related Works

**Robust MARL Strategies:** Recent research has focused on robust MARL to address unexpected changes in multi-agent environments. Max-min optimization (Chinchuluun et al., 2008; Han & Sung, 2021) has been applied to traditional MARL algorithms for robust learning (Li et al., 2019; Wang et al., 2022). Robust Nash equilibrium has been redefined to better suit multi-agent systems (Zhang et al., 2020b; Li et al., 2023a). Regularization-based approaches have also been explored to improve MARL robustness (Lin et al., 2020; Li et al., 2023b; Wang et al., 2023; Bukharin et al., 2024), alongside distributional reinforcement learning methods to manage uncertainties (Li et al., 2020; Xu et al., 2021; Du et al., 2024; Geng et al., 2024).

**Adversarial Attacks for Resilient RL:** To strengthen RL, numerous studies have explored adversarial learning to train policies under worst-case scenarios (Pattanaik et al., 2017; Tessler et al., 2019; Pinto et al., 2017; Chae et al., 2022). These attacks introduce perturbations to various MDP components, including state (Zhang et al., 2020a; 2021a; Everett et al., 2021; Li et al., 2023c; Qiaoben et al., 2024), action (Tan et al., 2020; Lee et al., 2021; Liu et al., 2024), and reward (Wang et al., 2020a; Zhang et al., 2020c; Rakhsha et al., 2021; Xu et al., 2022; Cai et al., 2023; Bouhaddi & Adi, 2023; Xu et al., 2024; Bouhaddi & Adi, 2024). Adversarial attacks have recently been extended to multi-agent setups, introducing uncertainties to state or observation (Han et al., 2022; He et al., 2023; Zhang et al., 2023; Zhou et al., 2023), actions (Yuan et al., 2023), and rewards (Kardeş et al., 2011). Further research has applied adversarial attacks to value decomposition frameworks (Phan et al., 2021), selected critical agents for targeted attacks (Yuan et al., 2023; Zhou et al., 2024), and analyzed their effects on inter-agent communication (Xue et al., 2021; Tu et al., 2021; Sun et al., 2023; Yuan et al., 2024).

**Model-based Frameworks for Robust RL:** Model-based methods have been extensively studied to enhance RL robustness (Berkenkamp et al., 2017; Panaganti & Kalathil, 2021; Curi et al., 2021; Clavier et al., 2023; Shi & Chi, 2024; Ramesh et al., 2024), including adversarial extensions

(Wang et al., 2020c; Kobayashi, 2024). Transition models have been leveraged to improve robustness (Mankowitz et al., 2019; Ye et al., 2024; Herremans et al., 2024), and offline setups have been explored for robust training (Rigter et al., 2022; Bhardwaj et al., 2024). In multi-agent systems, model-based approaches address challenges like constructing worst-case sets (Shi et al., 2024) and managing transition kernel uncertainty (He et al., 2022).

## 3. Background

### 3.1. Dec-POMDP and Value-based CTDE Setup

A fully cooperative multi-agent environment is modeled as a decentralized partially observable Markov decision process (Dec-POMDP) (Oliehoek et al., 2016), defined by the tuple $\mathcal{M} = \langle \mathcal{N}, \mathcal{S}, \mathcal{A}, P, \Omega, O, R, \gamma \rangle$. $\mathcal{N} = 1, \ldots, n$ is the set of agents, $\mathcal{S}$ the global state space, $\mathcal{A} = \mathcal{A}^1 \times \cdots \times \mathcal{A}^n$ the joint action space, $P$ the state transition probability, $\Omega$ the observation space, $R$ the reward function, and $\gamma \in [0, 1)$ the discount factor. At time $t$, each agent $i$ observes $o_t^i = O(s_t, i) \in \Omega$ and takes action $a_t^i \in \mathcal{A}^i$ based on its individual policy $\pi^i(\cdot | \tau_t^i)$, where $\tau_t^i$ is the agent's trajectory up to $t$. The joint action $\mathbf{a}_t = \langle a_t^1, \ldots, a_t^n \rangle$ sampled from the joint policy $\pi := \langle \pi^1, \cdots, \pi^n \rangle$ leads to the next state $s_{t+1} \sim P(\cdot | s_t, \mathbf{a}_t)$ and reward $r_t := R(s_t, \mathbf{a}_t)$. MARL aims to find the optimal joint policy that maximizes $\sum_{t=0}^{\infty} \gamma^t r_t$. As noted, this paper adopts the centralized training with decentralized execution (CTDE) setup, where the joint value $Q^{tot}(s_t, \mathbf{a}_t)$ is learned using temporal-difference (TD) learning. Through credit assignment, individual value functions $Q^i(\tau_t^i, a_t^i)$ are learned, guiding individual policies $\pi^i$ to select actions that maximize $Q^i$, i.e., $\pi^i := \arg\max_{a_t^i \in \mathcal{A}^i} Q^i(\tau_t^i, \cdot)$.

### 3.2. Robust MARL with Adversarial Attack Policy

Among various methods for robust learning in MARL, Yuan et al. (2023) defined an adversarial attack policy $\pi_{\text{adv}}$ and implemented robust MARL by training multi-agent policies to defend against attacks executed by $\pi_{\text{adv}} : \mathcal{S} \times \mathcal{A} \times \mathbb{N} \to \mathcal{A}$. A cooperative MARL environment with an adversarial policy $\pi_{\text{adv}}$ can be described as a Limited Policy Adversary Dec-POMDP (LPA-Dec-POMDP) $\tilde{\mathcal{M}}$, defined as follows:

**Definition 3.1** (Limited Policy Adversary Dec-POMDP). Given a Dec-POMDP $\mathcal{M}$ and a fixed adversarial policy $\pi_{\text{adv}}$, we define a Limited Policy Adversary Dec-POMDP (LPA-Dec-POMDP) $\tilde{\mathcal{M}} = \langle \mathcal{N}, \mathcal{S}, \mathcal{A}, P, K, \Omega, O, R, \gamma \rangle$, where $K$ is the maximum number of attacks, $\pi_{\text{adv}}(\cdot | s_t, \mathbf{a}_t, k_t)$ executes joint action $\tilde{\mathbf{a}}_t$ to disrupt the original action $\mathbf{a}_t$ chosen by $\pi$, and $k_t \leq K$ indicates the number of remaining attacks.

Here, if $\pi_{\text{adv}}$ selects an attack action $\tilde{\mathbf{a}}_t$ different from the original action $\mathbf{a}_t$, the remaining number of attacks $k_t$ de-

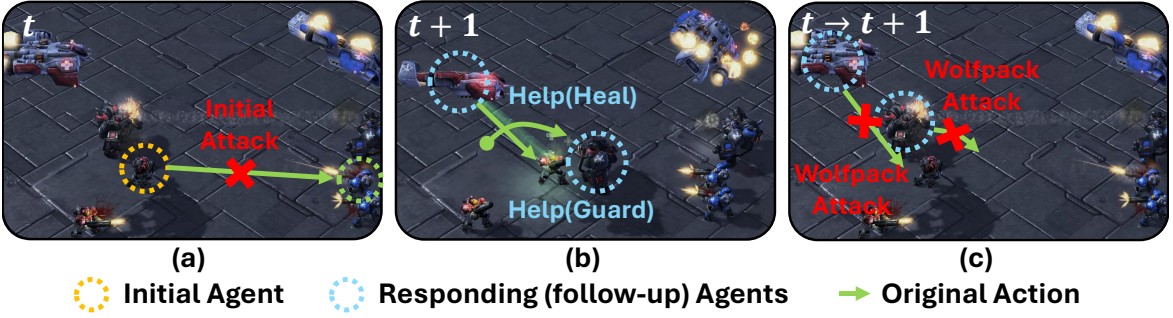

**(a)** ○ **Initial Agent**   **(b)** ○ **Responding (follow-up) Agents**   **(c)** → **Original Action**

*Figure 1.* Visualization of Wolfpack attack strategy during combat in the StarCraft II environment: (a) The initial agent is attacked, disrupting its original action (b) Responding (follow-up) agents to help the initially attacked agent and (c) Wolfpack adversarial attack that disrupts help actions of follow-up agents.

creases by 1. Once $k_t$ reaches 0, no further attacks can be performed. In this framework, Yuan et al. (2023) also demonstrated that the LPA-Dec-POMDP can be represented as another Dec-POMDP induced by $\pi_{\mathrm{adv}}$, with the convergence of the optimal policy $\pi^*$ under $\tilde{\mathcal{M}}$ guaranteed. In particular, it considers an adversarial attack targeting a chosen agent $i$ by minimizing its individual value function $Q^i$, as proposed by (Pattanaik et al., 2017), i.e., $\tilde{a}_t^i = \arg\min_{a \in \mathcal{A}^i} Q^i(\tau_t^i, a)$. Additionally, to enhance the attack's effectiveness, an evolutionary generation of attacker (EGA) approach is proposed, which combines multiple adversarial policies.

# 4. Methodology

## 4.1. Motivation of Wolfpack Attack Strategy

Existing adversarial attackers typically target only a single agent per attack, without coordination or relationships between successive attacks. In a cooperative MARL setup, such simplistic attacks enable non-targeted agents to learn effective policies to counteract the attack. However, we observe that policies trained under these conditions are vulnerable to coordinated attacks. As illustrated in Fig. 1(a), a single agent is attacked at time $t$. In Fig. 1(b), at the next step $t+1$, responding agents adjust their actions, such as healing or moving to guard, to protect the initially attacked agent. In contrast, Fig. 1(c) demonstrates a coordinated attack strategy that targets the agents responding to the initial attack. Such coordinated attacks render the learned policy ineffective, preventing it from countering the attacks entirely. This highlights that coordinated attacks are far more detrimental than existing attack methods, and current robust policies fail to defend effectively against them.

As depicted in Fig. 1(c), targeting agents that respond to an initial attack aligns with the Wolfpack attack strategy, a tactic widely employed in traditional military operations, as discussed in Section 1. To adapt this concept to a cooperative multi-agent setup, we define a Wolfpack adversarial attack as a coordinated strategy where one agent is attacked

initially, followed by targeting the group of follow-up agents that respond to defend against the initial attack, as shown in Fig. 1(c). Leveraging this approach, we aim to develop robust policies capable of effectively countering Wolfpack adversarial attack, thereby significantly enhancing the overall resilience of the learning process.

## 4.2. Wolfpack Adversarial Attack

In this section, we formally propose the Wolfpack adversarial attack, as introduced in the previous sections. The Wolfpack attack consists of two components: initial attacks, where a single agent is targeted at a specific time step $t_{\mathrm{init}}$, and follow-up group attacks, where the group of agents responding to the initial attack is selected and targeted over the subsequent steps $t_{\mathrm{init}} + 1, \cdots, t_{\mathrm{init}} + t_{\mathrm{WP}}$. Over the course of an episode, a maximum of $K_{\mathrm{WP}}$ Wolfpack attacks can be executed. Consequently, the total number of attack steps is given by $K = K_{\mathrm{WP}} \times (t_{\mathrm{WP}} + 1)$. The Wolfpack adversarial attacker $\pi_{\mathrm{adv}}^{\mathrm{WP}}$ can then be defined as follows:

**Definition 4.1** (Wolfpack Adversarial Attacker). A Wolfpack adversarial attacker $\pi_{\mathrm{adv}}^{\mathrm{WP}} : \mathcal{S} \times \mathcal{A} \times \mathbb{N} \to \mathcal{A}$ is defined as $\tilde{\mathbf{a}}_t = \pi_{\mathrm{adv}}^{\mathrm{WP}}(s_t, \mathbf{a}_t, k_t)$, where $\tilde{a}_t^i = \arg\min_{a_t^i \in \mathcal{A}^i} Q^{tot}(s_t, a_t^i, \mathbf{a}_t^{-i})$ for all $i \in \mathcal{N}_{t,\mathrm{attack}}$, and $\tilde{a}_t^i = a_t^i$ otherwise. Here, $\mathbf{a}_t^{-i}$ represents the joint actions of all agents excluding the $i$-th agent, and $\mathcal{N}_{t,\mathrm{attack}}$ denotes the set of agents targeted for adversarial attack, defined as

$$\mathcal{N}_{t,\mathrm{attack}} = \begin{cases} \emptyset & \text{if } k_t = 0, \\ \{i\} & \text{else if } t = t_{\mathrm{init}}, i \sim \mathrm{Unif}(\mathcal{N}), \\ \mathcal{N}_{\mathrm{follow-up}} & \text{else if } t = t_{\mathrm{init}} + 1, \cdots, t_{\mathrm{init}} + t_{\mathrm{WP}}, \\ \emptyset & \text{otherwise}, \end{cases}$$

where $\mathrm{Unif}(\cdot)$ is the Uniform distribution, $\mathcal{N}_{\mathrm{follow-up}} := \{i_1, \cdots, i_m\} \subset \mathcal{N}$ is the group of agents selected for follow-up attack, and $m$ is the number of follow-up agents.

Here, note that $k_t$ decreases by 1 for every attack step such that $\tilde{\mathbf{a}}_t \neq \mathbf{a}_t$, as in the ordinary adversarial attack policy, and the total value function $Q^{tot}$ is used for the attack instead

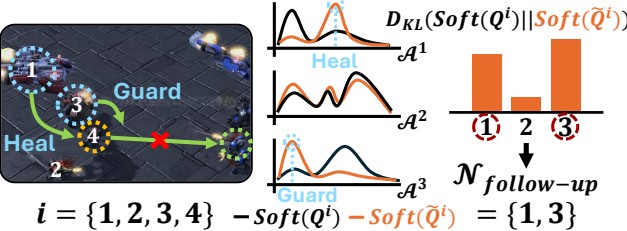

*Figure 2.* Visualization of follow-up agent group selection method: Agent 4 is initially attacked, and the $m$ agents exhibiting the largest changes in $Q^i$ are selected from $\{1, 2, 3\}$ ($m = 2$).

of $Q^i$. The proposed Wolfpack adversarial attacker $\pi_{\text{adv}}^{\text{WP}}$ is a special case of the adversarial policy defined in Definition 3.1. Consequently, the proposed attacker forms an LPA-Dec-POMDP $\tilde{\mathcal{M}}$ induced by $\pi_{\text{adv}}^{\text{WP}}$, and as demonstrated in Yuan et al. (2023), the convergence of MARL within the LPA-Dec-POMDP can be guaranteed. The proposed Wolfpack attack involves two key issues: how to design the group of follow-up agents $\mathcal{N}_{\text{follow-up}}$ and when to select $t_{\text{init}}$. The following sections address these aspects in detail.

### 4.3. Follow-up Agent Group Selection Method

In the Wolfpack adversarial attacker, we aim to identify the follow-up agent group $\mathcal{N}_{\text{follow-up}}$ that actively responds to the initial attack $\pi_{\text{adv}}^{\text{WP}}(s_{t_{\text{init}}}, \mathbf{a}_{t_{\text{init}}}, k_{t_{\text{init}}})$ and target them in subsequent steps. To do this, we define the difference between the $Q$-functions from the original action and the initial attack at time $t$ as:

$$\Delta Q_t^{tot} = Q^{tot}(s_t, \mathbf{a}_t) - Q^{tot}(s_t, \tilde{\mathbf{a}}_t),$$

where $\Delta Q_t^{tot} \geq 0$ for all $t$ such that $\mathcal{N}_{t,\text{attack}} \neq \emptyset$, because $\tilde{\mathbf{a}}_t$ minimizes $Q^{tot}$ for the agent indices selected by $\pi_{\text{adv}}^{\text{WP}}$. Assuming the $i$-th agent is the target of the initial attack, updating $Q^{tot}$ based on $\Delta Q_{t_{\text{init}}}^{tot}$ adjusts each agent's individual value function $Q^j$ to increase $Q^{tot}$ for all $j \neq i \in \mathcal{N}$, in accordance with the credit assignment principle in CTDE algorithms (Sunehag et al., 2017; Rashid et al., 2020), as shown below:

$$\tilde{Q}^i(\tau_{t_{\text{init}}}^i, \cdot) = Q^i(\tau_{t_{\text{init}}}^i, \cdot) - \alpha_{\text{lr}} \left. \frac{\partial \Delta Q_{t_{\text{init}}}^{tot}}{\partial Q^i(\tau_{t_{\text{init}}}^i, \mathbf{a})} \right|_{\mathbf{a} = \tilde{\mathbf{a}}_{t_{\text{init}}}}, \tag{1}$$

where $\alpha_{\text{lr}}$ is the learning rate. As agents select actions based on $Q^j$, changes in $Q^j$ indicate adjustments in their policies in response to the initial attack. Agents with the largest changes in $Q^j$ are identified as follow-up agents, while the $i$-th agent is excluded as it is already under attack and cannot respond immediately.

To identify the follow-up agent group, the updated $\tilde{Q}^j$ and original $Q^j$ are transformed into distributions using the Softmax function $\text{Soft}()$. This transformation softens the deterministic policy $\pi^j$, which directly selects an action to

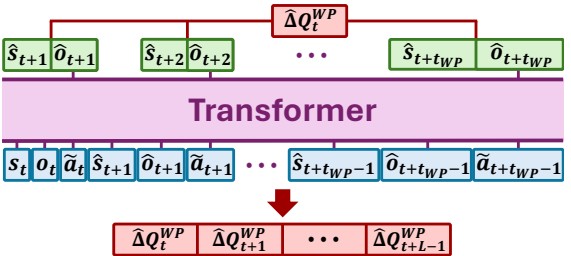

*Figure 3.* Planning with Transformer

maximize $Q^j$, making distributional differences easier to compute. The follow-up agent group is determined by selecting the $m$ agents that maximize the Kullback-Leibler (KL) divergence $D_{\text{KL}}$ between these distributions:

$$\mathcal{N}_{\text{follow-up}} = \operatorname*{arg\,max}_{\mathcal{N}' \subset \mathcal{N}, |\mathcal{N}'| = m, j \in \mathcal{N}', j \neq i} \sum_j D_{\text{KL}} \big( \tag{2}$$
$$\text{Soft}(Q^j(\tau_{t_{\text{init}}}^j, \cdot)) || \text{Soft}(\tilde{Q}^j(\tau_{t_{\text{init}}}^j, \cdot)) \big).$$

Using the proposed method, the follow-up agent group is identified as the agents whose policy distributions experience the most significant changes following the initial attack. Fig. 2 illustrates this process. After the initial attack, $Q$-differences are computed for the remaining agents $1, 2, 3$, and those with the largest changes in individual value functions are selected as the follow-up agent group. These agents are targeted over the next $t_{\text{WP}}$ time steps to prevent them from effectively responding. In Section 5, we analyze how the proposed method enhances attack criticalness by comparing it to naive selection methods based solely on observation distances.

### 4.4. Planner-based Critical Attacking Step Selection

In the proposed Wolfpack adversarial attacker $\pi_{\text{adv}}^{\text{WP}}$, the follow-up agent group is defined, leaving the task of determining the timing of initial attacks $t_{\text{init}}$, executed $K_{\text{WP}}$ times within an episode. While Random Step Selection involves choosing time steps randomly, existing methods show that selecting steps to minimize the rewards of the execution policy $\pi$ leads to more effective attacks and facilitates robust learning (Yuan et al., 2023). However, in coordinated attacks like Wolfpack, targeting steps that cause the greatest reduction in the Q-function value $\Delta Q_t^{\text{WP}}$ ensures a more devastating and lasting impact on the agents' ability to recover and respond. Thus, we propose selecting initial attack times based on the total reduction in $\Delta Q_t^{\text{WP}}$, defined as:

$$\Delta Q_t^{\text{WP}} = \sum_{l=t}^{t+t_{\text{WP}}} \Delta Q_l^{tot},$$

where the Wolfpack attack is performed from $t$ (initial attack) to $t + 1, \cdots, t + t_{\text{WP}}$ (follow-up attacks). Initial attack time steps $t_{\text{init}}$ are chosen to maximize $\Delta Q_t^{\text{WP}}$, which

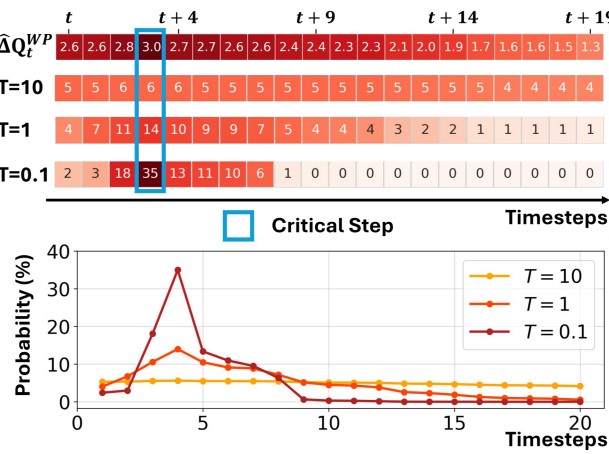

Figure 4. Attacking step probabilites

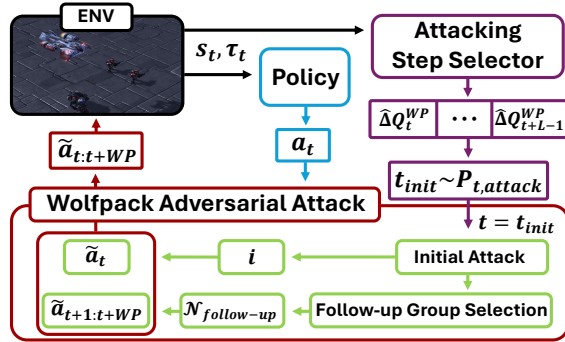

Figure 5. Illustration of the proposed WALL framework

captures the total $Q$-value reduction caused by the attack over $t_{\mathrm{WP}} + 1$ steps, enhancing the criticalness of the attack. However, computing $\Delta Q_t^{\mathrm{WP}}$ for every time step is computationally expensive as it requires generating attacked samples through interactions with the environment. To mitigate this, a stored buffer is utilized to plan trajectories of future states and observations for the attack.

For planning, we employ a Transformer (Vaswani, 2017), commonly used in sequential learning, which leverages an attention mechanism for efficient learning. As shown in Fig. 3, the Transformer learns environment dynamics $P$ using replay buffer trajectories to predict future states and observations $(\hat{s}_{t+1}, \hat{\mathbf{o}}_{t+1}, \cdots, \hat{s}_{t+t_{\mathrm{WP}}}, \hat{\mathbf{o}}_{t+t_{\mathrm{WP}}})$, where $\mathbf{o}_t$ represents the joint observation used to compute $Q^{tot}$. Actions $\tilde{\mathbf{a}}_l = \pi_{\mathrm{adv}}^{\mathrm{WP}}(\hat{s}_l, \mathbf{a}_l, k_l)$ for $\mathbf{a}_l \sim \pi$ are generated by $\pi_{\mathrm{adv}}^{\mathrm{WP}}$ for $l = t, \cdots, t + t_{\mathrm{WP}}$, with $\hat{s}_t = s_t$. Using the planner, we estimate the $Q$-value reduction $\hat{\Delta} Q_t^{\mathrm{WP}}$ caused by the Wolfpack attack. For $L$ time steps $l = t, \cdots, t + L - 1$, we compute future $Q$-differences $\hat{\Delta} Q_l^{\mathrm{WP}}$ and select $t_{\mathrm{init}}$ based on the initial attack probability $P_{t,\mathrm{attack}}$:

$$P_{t,\mathrm{attack}} = \left\{ \mathrm{Soft}\left( \hat{\Delta} Q_t^{\mathrm{WP}}/T, \cdots, \hat{\Delta} Q_{t+L-1}^{\mathrm{WP}}/T \right) \right\}_1, \tag{3}$$

where $\mathbf{x}_l$ indicates the $l$-th element of $\mathbf{x}$, and $T > 0$ is the temperature. In this paper, we set $L = 20$ as it provides an appropriate attack period. After selecting $K_{\mathrm{WP}}$ initial attacks, no further attacks are performed. Fig. 4 shows how step probabilities are distributed for different $T$ values ($T = 0.1, 1, 10$). At each time $t$, the planner predicts $\hat{\Delta} Q_t^{\mathrm{WP}}$ for $t$ to $t + L - 1$, forming soft initial attack probabilities. A larger $T$ results in more uniform probabilities, while a smaller $T$ increases the likelihood of targeting critical steps where $\hat{\Delta} Q_t^{\mathrm{WP}}$ is highest. These critical steps are selected with the highest probabilities for initial attacks. In Section 5, we analyze the effectiveness of this method in delivering more critical attacks compared to Random Step Selection and examine the impact of $T$ on performance in practical

environments. Since the proposed method involves planning at every evaluation, we also train a separate model to predict $\hat{\Delta} Q_t^{\mathrm{WP}}$, significantly reducing computational complexity. Details of this approach and the Transformer training loss functions are provided in Appendix B.1.

### 4.5. WALL: A Robust MARL Algorithm

Similar to other robust MARL methods, we propose the Wolfpack-Adversarial Learning for MARL (WALL) framework, a robust policy designed to counter the Wolfpack attack by performing MARL on the LPA-Dec-POMDP $\tilde{\mathcal{M}}$ with the Wolfpack attacker $\pi_{\mathrm{adv}}^{\mathrm{WP}}$. While the proposed Wolfpack framework is broadly applicable to most CTDE algorithms, we primarily applied it to well-known value-based CTDE methods, including QMIX (Rashid et al., 2020), VDN (Sunehag et al., 2017), and QPLEX (Wang et al., 2020b). Detailed implementations, including loss functions for the planner Transformer and the value functions, are provided in Appendix B.2. The proposed WALL framework is illustrated in Fig. 5 and summarized in Algorithm 1.

---

**Algorithm 1** WALL framework

---

1: **Initialize:** Value function $Q^{tot}$, Planning Transformer
2: **for** each training iteration **do**
3:     **for** each environment step $t$ **do**
4:         Sample the action $\mathbf{a}_t$: $a_t^i \sim \epsilon\text{-greedy}(Q^i)$
5:         Compute $P_{t,\mathrm{attack}}$ using Planner and sample $t_{\mathrm{init}}$
6:         **if** $t = t_{\mathrm{init}}$ **then**
7:             Perform the initial attack: $\tilde{\mathbf{a}}_t \sim \pi_{\mathrm{adv}}^{\mathrm{WP}}$
8:         **else if** $t_{\mathrm{init}} + 1 \leq t \leq t_{\mathrm{init}} + t_{\mathrm{WP}}$ **then**
9:             Select the follow-up agent group $\mathcal{N}_{\mathrm{follow-up}}$
10:        Perform the follow-up attack: $\tilde{\mathbf{a}}_t \sim \pi_{\mathrm{adv}}^{\mathrm{WP}}$
11:         **else**
12:             Execute the original action $\mathbf{a}_t$
13:         **end if**
14:     **end for**
15:     Update the $Q^{tot}$ using a CTDE algorithm
16:     Update the Planning Transformer
17: **end for**

---

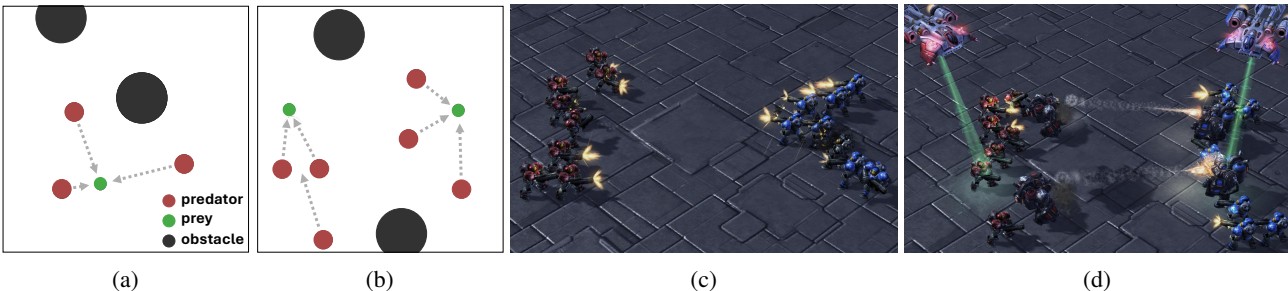

*Figure 6.* MARL benchmarks used in our experiments: (a) PP_3/1 and (b) PP_6/2 in MPE, and (c) 8m and (d) MMM scenarios in SMAC.

# 5. Experiments

In this section, we evaluate the proposed methods on two standard benchmarks in MARL research: the Multi-Agent Particle Environment (MPE) (Lowe et al., 2017) and the StarCraft II Multi-Agent Challenge (SMAC) (Samvelyan et al., 2019), as illustrated in Fig. 6. Specifically, we compare: (1) the impact of the proposed Wolfpack adversarial attack against other adversarial attacks, and (2) the robustness of the WALL framework in defending against such attacks compared to other robust MARL methods. Also, an ablation study analyzes the effect of the proposed components and hyperparameters on robustness. All results are reported as the mean and standard deviation (shaded areas for graphs and $\pm$ values for tables) across 5 random seeds. Our code is available at https://github.com/sunwoolee0504/WALL.

## 5.1. Environmental Setup

The MPE environment provides a multi-agent setting where agents interact through simple physical dynamics. We conduct experiments on three predator-prey (PP) scenarios with varying agent-to-target ratios: PP_3/1, PP_6/2, and PP_9/3. In these tasks, multiple predator agents must coordinate to capture one or more prey agents moving adversarially. The SMAC environment serves as a challenging benchmark requiring effective agent cooperation to defeat opponents. We evaluate the proposed method across six scenarios: 2s3z, 3m, 3s_vs_3z, 8m, MMM, and 1c3s5z. We perform parameter searches for the number of follow-up agents $m$, total Wolfpack attacks $K_{\mathrm{WP}}$, and attack duration $t_{\mathrm{WP}}$, using optimal settings for comparisons. To ensure realistic constraints, we set $m$ to $m < \left\lfloor \frac{n-1}{2} \right\rfloor$, where $n$ is the maximum number of allied units. We provide details on environment setups and experimental configurations, including hyperparameter settings, in Appendices A and C. All MARL methods are evaluated on the QMIX baseline, with comparison for other CTDE baselines in Appendix E.1.

**Adversarial Attacker Baselines:** To compare the severity of different attacks, we consider the following 4 scenarios: **Natural**, representing the case where no attacks are per-

| Method | Scenario | PP_3/1 | PP_6/2 | PP_9/3 | Mean |
|---|---|---|---|---|---|
| Natural | Vanilla QMIX | 165.4 ± 1.3 | 538.8 ± 5.5 | 661.9 ± 5.4 | 455.4 ± 2.1 |
| | RANDOM | 178.3 ± 1.0 | 663.8 ± 4.0 | 666.9 ± 3.5 | 503.0 ± 1.3 |
| | ROMANCE | 175.0 ± 0.7 | 648.6 ± 3.9 | 721.4 ± 7.5 | 515.0 ± 1.4 |
| | WALL (ours) | **202.9 ± 1.7** | **675.0 ± 3.8** | **802.5 ± 5.7** | **560.1 ± 1.7** |
| Random Attack | Vanilla QMIX | 158.9 ± 2.1 | 522.6 ± 2.4 | 656.4 ± 3.0 | 445.9 ± 2.0 |
| | RANDOM | 170.1 ± 1.3 | 638.7 ± 2.0 | 657.9 ± 2.2 | 488.9 ± 1.9 |
| | ROMANCE | 173.4 ± 1.4 | 624.2 ± 3.5 | 701.7 ± 4.0 | 499.7 ± 2.4 |
| | WALL (ours) | **202.9 ± 1.7** | **654.0 ± 2.1** | **802.5 ± 2.1** | **553.1 ± 1.7** |
| EGA | Vanilla QMIX | 153.7 ± 2.0 | 499.8 ± 0.6 | 585.1 ± 1.7 | 412.8 ± 1.4 |
| | RANDOM | 157.6 ± 1.9 | 604.6 ± 3.2 | 589.7 ± 1.7 | 450.6 ± 1.4 |
| | ROMANCE | 167.1 ± 2.4 | 605.1 ± 2.7 | 594.4 ± 1.3 | 455.5 ± 1.5 |
| | WALL (ours) | **166.5 ± 1.7** | **685.5 ± 5.0** | **682.7 ± 0.8** | **511.5 ± 1.4** |
| Wolfpack Adversarial Attack (ours) | Vanilla QMIX | 135.7 ± 1.5 | 415.6 ± 8.4 | 551.7 ± 7.0 | 367.6 ± 5.0 |
| | RANDOM | 157.5 ± 1.4 | 571.0 ± 9.0 | 624.3 ± 7.7 | 450.9 ± 8.4 |
| | ROMANCE | 159.3 ± 1.5 | 554.6 ± 9.9 | 570.3 ± 7.5 | 428.0 ± 9.4 |
| | WALL (ours) | **171.5 ± 1.5** | **599.0 ± 9.4** | **698.1 ± 9.1** | **489.5 ± 0.6** |

*Table 1.* Average cumulative rewards of robust MARL policies under various attack settings in the MPE environments.

formed; **Random Attack**, where time steps, agents, and actions are randomly selected to execute attacks; **Evolutionary Generation of Attackers (EGA)** (Yuan et al., 2023), which combines multiple single-agent-targeted attackers generated from various seeds as described in Section 3; and the proposed **Wolfpack Adversarial Attack**. For a fair comparison, adversarial attackers are trained on independent seeds to execute unseen attacks.

**Robust MARL Baselines:** To compare the severity of attack baselines and the robustness of policies trained under adversarial attack scenarios, we evaluate QMIX-trained policies under the following attack conditions: **Vanilla QMIX**, assuming no adversarial attacks; **RANDOM**, using Random Attack; **RARL** (Pinto et al., 2017), where adversarial attackers tailor attacks to the learned policy; **RAP** (Vinitsky et al., 2020), an extension of RARL that uniformly samples attackers to prevent overfitting and introduce diversity; **ROMANCE** (Yuan et al., 2023), an RAP extension countering diverse EGA attacks; **ERNIE** (Bukharin et al., 2024), enhancing robustness via adversarial regularization in observations and actions; and the proposed **WALL**. All robust MARL methods follow author-provided methodologies and parameters. Further details on the MARL baselines are available in Appendix D. All policies are trained for 3M timesteps, starting from a pretrained Vanilla QMIX model trained for 1M timesteps.

| | Scenario / Method | 2s3z | 3m | 3s_vs_3z | 8m | MMM | 1c3s5z | Mean |
|---|---|---|---|---|---|---|---|---|
| Natural | Vanilla QMIX | $98.0 \pm 1.5$ | $99.2 \pm 1.0$ | $99.2 \pm 1.6$ | $97.6 \pm 2.1$ | $99.2 \pm 0.5$ | $99.1 \pm 1.1$ | $98.7 \pm 0.5$ |
| | RANDOM | $\mathbf{99.7 \pm 0.5}$ | $99.1 \pm 1.2$ | $99.0 \pm 0.8$ | $99.2 \pm 1.0$ | $\mathbf{99.6 \pm 0.6}$ | $99.3 \pm 1.0$ | $99.3 \pm 0.2$ |
| | RARL | $97.8 \pm 2.0$ | $93.8 \pm 3.2$ | $93.1 \pm 17.4$ | $95.5 \pm 3.7$ | $90.6 \pm 20.8$ | $84.0 \pm 33.7$ | $92.5 \pm 5.3$ |
| | RAP | $98.8 \pm 1.3$ | $95.8 \pm 4.4$ | $99.5 \pm 1.0$ | $94.7 \pm 6.7$ | $95.5 \pm 12.1$ | $84.2 \pm 16.9$ | $94.7 \pm 2.6$ |
| | ERNIE | $98.2 \pm 1.3$ | $99.2 \pm 1.2$ | $99.8 \pm 0.4$ | $\mathbf{99.8 \pm 0.5}$ | $98.5 \pm 1.7$ | $99.2 \pm 1.0$ | $99.1 \pm 0.5$ |
| | ROMANCE | $96.4 \pm 2.9$ | $93.6 \pm 13.7$ | $99.7 \pm 0.5$ | $99.6 \pm 0.6$ | $96.4 \pm 6.6$ | $96.5 \pm 4.3$ | $97.0 \pm 2.1$ |
| | WALL (ours) | $99.4 \pm 0.6$ | $\mathbf{99.7 \pm 0.8}$ | $\mathbf{99.8 \pm 0.7}$ | $99.3 \pm 0.6$ | $99.0 \pm 2.1$ | $\mathbf{99.5 \pm 0.6}$ | $\mathbf{99.4 \pm 0.5}$ |
| Random Attack | Vanilla QMIX | $80.4 \pm 3.2$ | $69.6 \pm 10.4$ | $91.4 \pm 4.6$ | $69.0 \pm 7.1$ | $66.4 \pm 20.5$ | $94.8 \pm 3.4$ | $78.6 \pm 2.2$ |
| | RANDOM | $90.8 \pm 3.8$ | $76.4 \pm 17.3$ | $97.4 \pm 0.6$ | $80.2 \pm 4.9$ | $95.4 \pm 5.3$ | $96.0 \pm 2.9$ | $89.4 \pm 2.5$ |
| | RARL | $86.8 \pm 4.0$ | $58.7 \pm 15.4$ | $89.2 \pm 20.4$ | $70.2 \pm 5.8$ | $84.2 \pm 5.4$ | $79.1 \pm 22.2$ | $78.0 \pm 7.2$ |
| | RAP | $91.0 \pm 4.7$ | $69.2 \pm 11.1$ | $97.8 \pm 1.6$ | $85.0 \pm 11.4$ | $86.7 \pm 30.3$ | $86.6 \pm 12.5$ | $83.0 \pm 3.7$ |
| | ERNIE | $83.2 \pm 6.9$ | $65.2 \pm 4.9$ | $90.2 \pm 9.2$ | $76.2 \pm 11.8$ | $86.0 \pm 18.2$ | $95.6 \pm 3.2$ | $82.7 \pm 5.2$ |
| | ROMANCE | $90.2 \pm 2.3$ | $71.6 \pm 10.8$ | $99.6 \pm 0.6$ | $84.8 \pm 5.0$ | $86.8 \pm 16.3$ | $94.0 \pm 1.9$ | $87.8 \pm 2.4$ |
| | WALL (ours) | $\mathbf{94.6 \pm 4.5}$ | $\mathbf{87.4 \pm 1.8}$ | $\mathbf{99.8 \pm 0.5}$ | $\mathbf{95.8 \pm 3.4}$ | $\mathbf{99.4 \pm 1.1}$ | $\mathbf{98.6 \pm 1.6}$ | $\mathbf{95.9 \pm 0.5}$ |
| EGA | Vanilla QMIX | $54.0 \pm 7.6$ | $66.5 \pm 15.5$ | $72.4 \pm 15.1$ | $71.2 \pm 20.0$ | $70.6 \pm 14$ | $83.0 \pm 2.6$ | $69.6 \pm 3.8$ |
| | RANDOM | $65.3 \pm 3.3$ | $70.6 \pm 38.6$ | $68.8 \pm 23.5$ | $87.5 \pm 5.5$ | $84.4 \pm 3.1$ | $84.5 \pm 2.9$ | $76.9 \pm 5.4$ |
| | RARL | $62.6 \pm 9.4$ | $74.4 \pm 12.4$ | $88.4 \pm 17.7$ | $78.4 \pm 9.1$ | $83.4 \pm 11.9$ | $80.1 \pm 11.4$ | $77.9 \pm 10.0$ |
| | RAP | $70.4 \pm 13.0$ | $84.4 \pm 7.3$ | $83.8 \pm 15.8$ | $86.2 \pm 3.8$ | $83.9 \pm 16.4$ | $80.2 \pm 5.4$ | $81.5 \pm 4.3$ |
| | ERNIE | $52.4 \pm 9.6$ | $60.4 \pm 20.1$ | $83.2 \pm 9.7$ | $81.6 \pm 8.5$ | $85.0 \pm 4.6$ | $93.6 \pm 2.2$ | $76.0 \pm 3.0$ |
| | ROMANCE | $79.8 \pm 2.8$ | $85.8 \pm 4.6$ | $91.0 \pm 5.1$ | $90.9 \pm 4.0$ | $87.8 \pm 11.7$ | $89.6 \pm 2.9$ | $87.5 \pm 1.6$ |
| | WALL (ours) | $\mathbf{88.6 \pm 5.4}$ | $\mathbf{87.0 \pm 5.4}$ | $\mathbf{98.7 \pm 0.8}$ | $\mathbf{95.8 \pm 2.9}$ | $\mathbf{94.2 \pm 3.8}$ | $\mathbf{97.0 \pm 1.3}$ | $\mathbf{93.6 \pm 1.5}$ |
| Wolfpack Adversarial Attack (ours) | Vanilla QMIX | $39.8 \pm 7.5$ | $31.0 \pm 11.8$ | $84.4 \pm 4.7$ | $11.4 \pm 13.3$ | $10.4 \pm 14.3$ | $59.2 \pm 4.2$ | $39.4 \pm 4.5$ |
| | RANDOM | $60.4 \pm 27.4$ | $57.4 \pm 30.5$ | $91.0 \pm 3.1$ | $40.4 \pm 14.8$ | $63.6 \pm 28.7$ | $68.4 \pm 19.6$ | $63.5 \pm 3.1$ |
| | RARL | $52.4 \pm 15.8$ | $31.1 \pm 20.3$ | $90.0 \pm 17.4$ | $14.2 \pm 9.7$ | $51.1 \pm 36.8$ | $75.9 \pm 13.6$ | $52.5 \pm 8.0$ |
| | RAP | $60.0 \pm 10.3$ | $37.5 \pm 10.4$ | $95.4 \pm 3.9$ | $35.6 \pm 14.4$ | $47.0 \pm 36.9$ | $75.7 \pm 26.1$ | $58.5 \pm 5.7$ |
| | ERNIE | $43.2 \pm 13.0$ | $35.4 \pm 6.2$ | $94.8 \pm 4.4$ | $26.4 \pm 10.4$ | $26.2 \pm 17.3$ | $77.0 \pm 9.3$ | $50.5 \pm 6.6$ |
| | ROMANCE | $62.4 \pm 5.1$ | $34.8 \pm 14.3$ | $98.6 \pm 0.6$ | $28.6 \pm 14.2$ | $48.8 \pm 17.9$ | $81.2 \pm 4.5$ | $59.1 \pm 2.0$ |
| | WALL (ours) | $\mathbf{92.2 \pm 3.7}$ | $\mathbf{90.8 \pm 4.9}$ | $\mathbf{99.8 \pm 0.5}$ | $\mathbf{83.6 \pm 5.0}$ | $\mathbf{95.0 \pm 4.5}$ | $\mathbf{98.8 \pm 1.6}$ | $\mathbf{93.4 \pm 1.1}$ |

*Table 2.* Average test win rates of robust MARL policies under various attack settings in the SMAC environments.

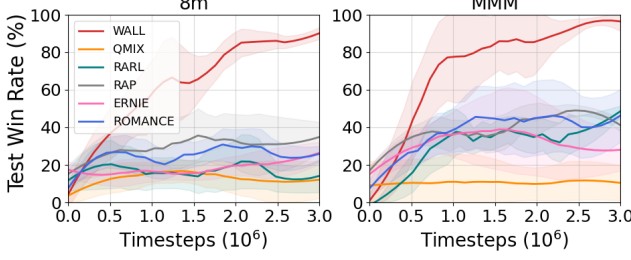

*Figure 7.* Learning curves of MARL methods for Wolfpack attack

### 5.2. Performance Comparison in MPE and SMAC

Table 1 presents the average cumulative rewards over the last 100 episodes under various attack settings in the MPE environments. The results show that the proposed Wolfpack adversarial attack is significantly more disruptive than existing methods such as EGA and Random Attack. For example, for Vanilla QMIX, the average cumulative reward across the three predator-prey scenarios drops by $455.4 - 367.6 = 87.8$ under the proposed Wolfpack attack, compared to $455.4 - 412.8 = 42.6$ under EGA and $455.4 - 445.9 = 9.5$ under Random Attack. These results demonstrate that the Wolfpack attack imposes a much more severe degradation in policy performance. In contrast, the proposed WALL framework consistently achieves the best performance across all attack types. Notably, in the MPE scenarios, WALL outperforms all baselines not only under adversarial attacks but also in the natural setting, demonstrat-

ing superior policy quality even without external threats.

For SMAC, Table 2 presents the average win rates over the last 100 episodes of MARL policies under different attack baselines. The results show that the proposed Wolfpack adversarial attack is significantly more powerful than existing methods such as EGA and Random Attack. For example, EGA reduces the performance of Vanilla QMIX by $98.7 - 69.6 = 29.1\%$ and RANDOM by $99.3 - 76.9 = 22.4\%$ compared to the natural scenario. In contrast, the Wolfpack attack reduces Vanilla QMIX performance by $98.7 - 39.4 = 59.3\%$ and RANDOM by $99.3 - 63.5 = 35.8\%$, demonstrating its greater impact. In addition, the proposed WALL framework, which is trained to defend against the Wolfpack attack, outperforms other robust MARL methods under all attack types, showcasing its superior robustness. Notably, although RANDOM is trained specifically against Random Attack and ROMANCE against EGA, WALL achieves better performance against both attack types. These results highlight the effectiveness of WALL in enabling robust learning under diverse adversarial scenarios. Fig. 7 further illustrates this in the 8m and MMM environments, where performance differences with existing methods are most pronounced, showing the average win rate of each policy over training steps under unseen Wolfpack adversarial attacks. The results reveal that WALL not only achieves higher robustness but also adapts more quickly to attacks. Similar trends are observed for other CTDE algo-

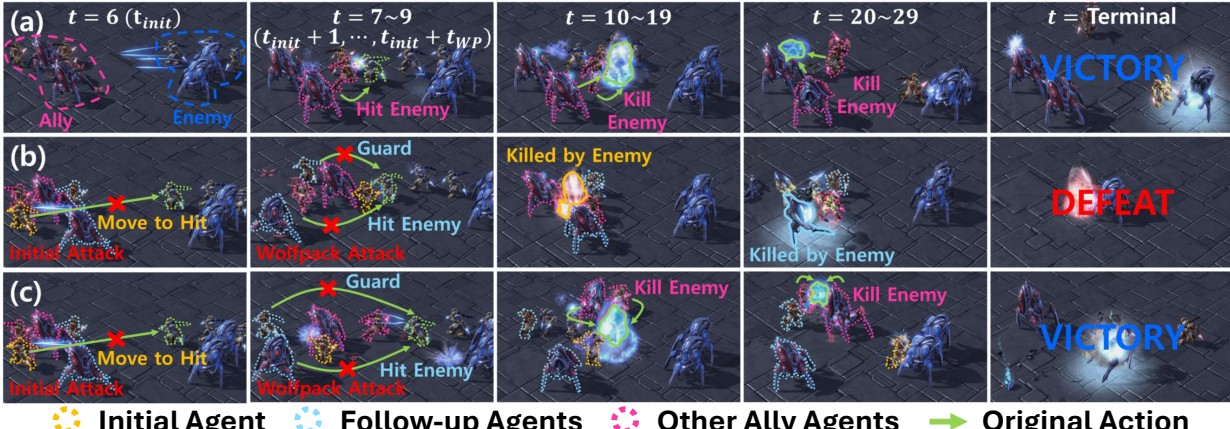

*Figure 8.* Attack comparison on 2s3z task in the SMAC: (a) QMIX/Natural, (b) QMIX/Wolfpack attack, and (c) WALL/Wolfpack attack

rithms, such as VDN and QPLEX, as detailed in Appendix E.1, confirming the robustness of the proposed method.

To support a more practical evaluation, we assess computational complexity and general robustness under common perturbations in Appendix E.4 and Appendix E.5, respectively. WALL incurs about 30% higher training cost than ROMANCE but achieves substantially better performance due to its critical step selection. For robustness, we consider perturbations including Gaussian noise in observations and test-time parameter shifts such as reduced allied HP, under which WALL still outperforms existing baselines. These results demonstrate the practical effectiveness of WALL under both computational and environmental challenges.

### 5.3. Visualization of Wolfpack Adversarial Attack

To analyze the superior performance of the Wolfpack attack, we provide a visualization of its execution in the SMAC environment. Fig. 8 illustrates a scenario where the proposed step selector identifies $t = 6$ as a critical initial step to initiate the attack. Prior to $t = 6$, all setups are assumed to follow the same trajectory. Fig. 8(a) shows Vanilla QMIX in a Natural scenario without attack, where our agents successfully defeat all enemy agents, achieving victory. Fig. 8(b) demonstrates Vanilla QMIX under the Wolfpack adversarial attack, with follow-up agents targeted during $t = 7$ to $t = 9$. This leaves other agents unable to effectively defend against the adversarial attack, resulting in defeat as all agents are killed by the enemy. Fig. 8(c) highlights a policy trained with the WALL framework. Despite the same follow-up agents are targeted during $t = 7$ to $t = 9$, WALL trains non-attacked agents to back up and protect the attacked agents, enabling ally agents to eliminate enemy agents and secure victory. This visualization demonstrates how the Wolfpack attack disrupts agent coordination and how the WALL framework robustly defends against such attacks. Visualizations of other SMAC tasks and detailed follow-up agent selection are provided in Appendix G.

### 5.4. Ablation Study

To evaluate the impact of each component and hyperparameter in the proposed Wolfpack adversarial attack, we conduct an ablation study focusing on the following aspects: component evaluation, step selection temperature $T$, and the number of follow-up agents $m$. The ablation study is conducted in the 8m and MMM environments, where the performance differences are most pronounced. Additionally, more ablation studies on other hyperparameters, such as the total number of Wolfpack attacks $K_{\mathrm{WP}}$ and the attack duration $t_{\mathrm{WP}}$, are provided in Appendix F.

**Component Evaluation:** To evaluate the impact of each proposed component on attack severity and policy robustness, we consider five setups: 'Default', which uses all proposed components as designed; 'Init. agent (min)', where the initial target agent $i$ is selected to minimize $Q^{tot}$, i.e., $i = \arg\min_j \min_{a^j} Q^{tot}(s_{t_{\mathrm{init}}}, a^j_{t_{\mathrm{init}}}, \mathbf{a}^{-j}_{t_{\mathrm{init}}})$, instead of random selection; 'Follow-up (L2)', which selects $m$ agents closest to the initial agent based on L2 distance instead of the proposed follow-up selection method; 'Step (Random)', which uses random step selection instead of the proposed step selection method, while keeping the same total number of attacks; and 'Agents & Step (Random)', which randomly selects both $m$ follow-up agents and attack steps.

For each setup, we train the Wolfpack adversarial attack and the corresponding robust policy of WALL. Fig. 9(a) shows the robustness of WALL trained under each setup when exposed to the default Wolfpack attack, while Fig. 9(b) illustrates how each attack setup degrades the performance of 'Vanilla QMIX' compared to its 'Natural' performance. Randomly selecting the initial agent yields more robust policies than selecting the agent minimizing $Q^{tot}$ ('Init. agent (min)'), as random selection introduces diversity in attack scenarios. While selecting the $Q^{tot}$-minimizing agent may slightly enhance attack severity in cases like MMM, the added diversity from random selection generally improves robust-

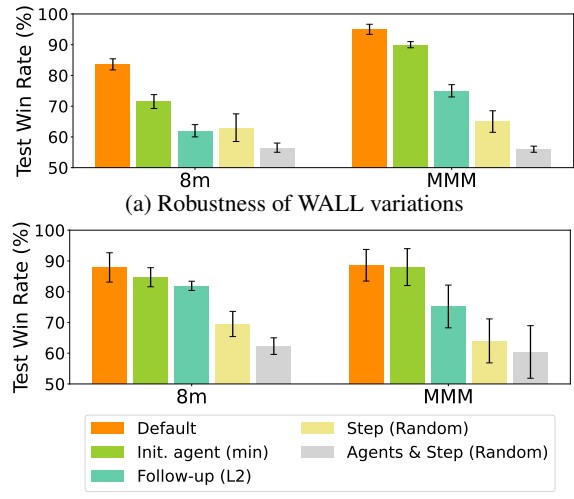

(a) Robustness of WALL variations

(b) Degradation of Vanilla QMIX under each attack setup

*Figure 9.* Component evaluation of our WALL/Wolfpack attack

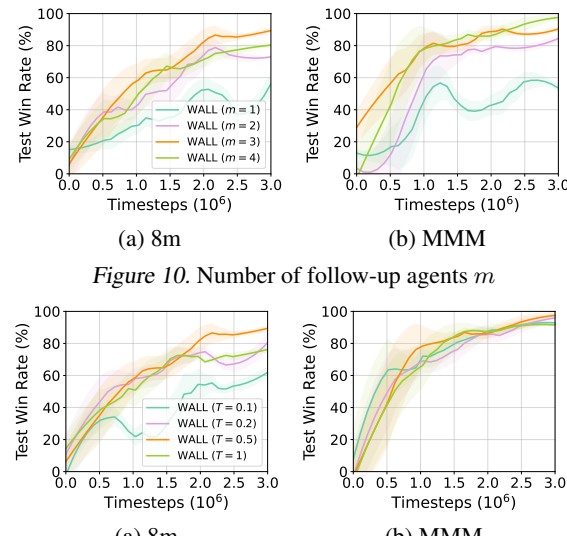

(a) 8m  (b) MMM

*Figure 10.* Number of follow-up agents $m$

(a) 8m  (b) MMM

*Figure 11.* Step selection temperature $T$

ness. Comparing 'Default' and 'Follow-up (L2)' shows that the proposed follow-up selection method enables more severe attacks and trains more robust policies than simply targeting agents closest to the initial agent. Similarly, 'Default' outperforms 'Step (Random)' in both attack severity and robustness, demonstrating that the proposed planner effectively identifies critical steps to minimize $Q^{tot}$, producing stronger policies. Finally, 'Default' achieves significantly better robustness and more critical attacks compared to 'Agents & Step (Random)', highlighting the combined effectiveness of the proposed components.

**Number of Follow-up Agents** $m$**:** To analyze the impact of the hyperparameter $m$, which determines the number of follow-up agents, on robustness, Fig. 10 shows how WALL trained with different values of $m$ defend against the default Wolfpack attack. To prevent excessive attack that could cause learning to fail, we assume $m < \left\lfloor \frac{n-1}{2} \right\rfloor$. The results indicate that in the 8m environment, when $m$ is small, only a few agents defending against the initial attack are targeted, leading to reduced robustness. Conversely, when $m = 4$, too many agents are attacked, causing learning to deteriorate. Therefore, $m = 3$ yields the most robust performance and is considered the default hyperparameter. Similarly, in the MMM environment, $m = 4$ results in the most robust performance and is set as the default. Notably, when $m = 1$, the attack becomes a single-agent attack. As discussed in Section 4.1, performing coordinated multi-agent attack ($m > 1$) enables much more robust learning, demonstrating the effectiveness and superiority of the proposed Wolfpack attack framework.

**Step Selection Temperature** $T$**:** $T$ is a hyperparameter in Eq. (3) that controls the temperature of the initial attack probability. A larger $T$ results in more random attacks across steps, while a smaller $T$ focuses attacks on critical steps with higher probability. Fig. 11 illustrates the perfor-

mance of WALL policies trained with varying values of $T$ against the default Wolfpack attack. In both the 8m and MMM environments, a very small $T$ causes the attack to target only specific steps, leading to policies that are less robust against diverse attacks. Conversely, a very large $T$ leads to overly uniform attacks, failing to target critical steps effectively, which also results in less robust policies. Based on these findings, we determined that $T = 0.5$ strikes a balance between targeting critical steps and maintaining robustness, and we set this as the default parameter.

## 6. Limitations

While the proposed WALL significantly improves robustness in MARL, it has a few limitations. The first is the additional computational overhead introduced by training the Transformer for identifying critical steps. However, as shown in our analysis, this overhead is justified given that other baselines fail to achieve comparable performance even with extended training. Another limitation is the need for hyperparameter tuning to construct the Wolfpack attack. Nevertheless, the method is not highly sensitive to these hyperparameters, and the provided ablation study offers practical guidelines for selecting appropriate configurations.

## 7. Conclusions

In this paper, we propose the Wolfpack adversarial attack, a coordinated strategy inspired by the Wolfpack tactic used in military operations, which significantly outperforms existing adversarial attacks. Additionally, we develop WALL, a robust MARL method designed to counter the proposed attack, demonstrating superior performance across various SMAC environments. Overall, our WALL framework enhances the robustness of MARL algorithms.

## Acknowledgment

This work was supported by Institute of Information & Communications Technology Planning & Evaluation (IITP) grant funded by the Korea government (MSIT) (No.RS-2022-II220469, Development of Core Technologies for Task-oriented Reinforcement Learning for Commercialization of Autonomous Drones), Innovative Human Resource Development for Local Intellectualization program through the Institute of Information & Communications Technology Planning & Evaluation (IITP) grant funded by the Korea government (MSIT) (IITP-2025-RS-2022-00156361), and Institute of Information & Communications Technology Planning & Evaluation (IITP) grant funded by the Korea government (MSIT) (No.RS-2020-II201336, Artificial Intelligence graduate school support (UNIST)).

## Impact Statement

This paper presents work whose goal is to advance the field of Machine Learning. There are many potential societal consequences of our work, none which we feel must be specifically highlighted here.

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

# A. Environmental Setup

We conduct experiments in the MPE (Lowe et al., 2017) and SMAC (Samvelyan et al., 2019) environments. This section provides detailed descriptions of their setup and features.

## A.1. Multi-Agent Particle Environments (MPE)

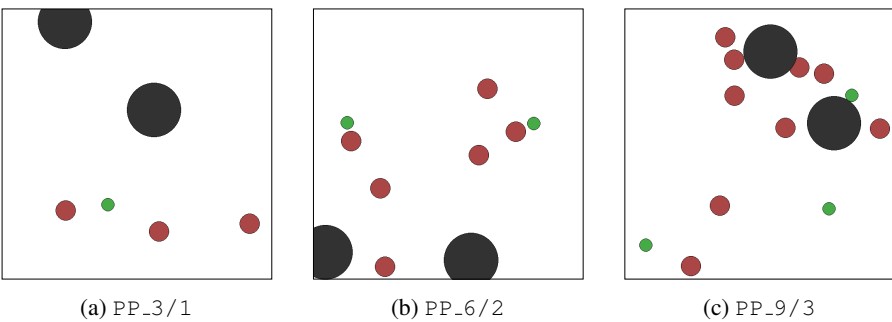

(a) PP_3/1         (b) PP_6/2         (c) PP_9/3

*Figure A.1.* Visualization of PP scenarios in the MPE environment

The Multi-Agent Particle Environment (MPE) (Lowe et al., 2017) is a widely used benchmark suite consisting of multi-agent scenarios. Agents are modeled as particles capable of movement and interaction, governed by simple physical dynamics. MPE includes both cooperative and competitive tasks, with each scenario sharing a continuous state space and typically partial observability. A standardized implementation of MPE is available through the PettingZoo library (Terry et al., 2021).

**Scenarios**
The MPE benchmark includes a variety of multi-agent scenarios. Among them, we focus on the predator-prey environment, which is well-suited for analyzing the impact of attacks on the cooperative structures among agents. We consider multiple variants denoted as PP_$X/Y$, where $X$ represents the number of predator agents and $Y$ the number of prey agents. In all scenarios, prey agents follow a random policy. Detailed configurations for the three selected variants are summarized in Table A.1 and Fig. A.1.

**State and observation spaces**
Each agent in the MPE environment receives a partial observation, which includes its own position and velocity, the relative positions and velocities of other predators, and the relative positions of landmarks. The global state is constructed by concatenating the local observations of all agents.

**Action space**
The action space is discrete. Each agent can choose one of the four cardinal directions or do nothing.

**Reward function**
The reward function $R$ assigns a positive value to predator agents upon a successful collision with a prey:

$$R = \sum_{g \in \text{prey}} \mathbb{I}(\text{collision}(g, \text{predator}))$$

where $\mathbb{I}(\cdot)$ is the indicator function, returning 1 if the condition inside is true and 0 otherwise. Here, collision$(i, j)$ denotes whether agent $i$ and agent $j$ are in physical contact.

| Map | Predators | Prey | State Dimension | Obs Dimension | Num. of Actions |
|---|---|---|---|---|---|
| PP_3/1 | 3 Agents | 1 Agent | 48 | 16 | 5 |
| PP_6/2 | 6 Agents | 2 Agents | 156 | 26 | 5 |
| PP_9/3 | 9 Agents | 3 Agents | 324 | 36 | 5 |

*Table A.1.* The number of agents, the dimensions of state and observation spaces, and the number of actions in MPE scenarios

## A.2. The StarCraft Multi-Agent Challenge (SMAC)

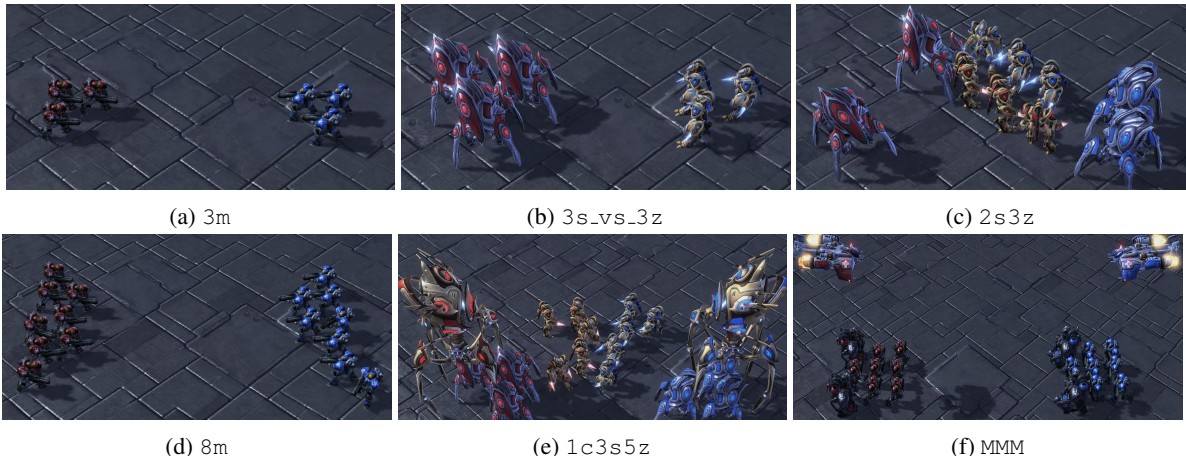

(a) `3m`                (b) `3s_vs_3z`                (c) `2s3z`

(d) `8m`                (e) `1c3s5z`                (f) `MMM`

*Figure A.2.* Visualization of SMAC scenarios

The StarCraft Multi-Agent Challenge (SMAC) serves as a benchmark for cooperative Multi-Agent Reinforcement Learning (MARL), focusing on decentralized micromanagement tasks. Based on the real-time strategy game StarCraft II, SMAC requires each unit to be controlled by independent agents acting solely on local observations. It offers a variety of combat scenarios to evaluate MARL methods effectively.

**Scenarios**
SMAC scenarios involve combat situations between an allied team controlled by learning agents and an enemy team managed by a scripted AI. These scenarios vary in complexity, unit composition, and terrain, challenging agents to use advanced micromanagement techniques such as focus fire, kiting, and terrain exploitation. Scenarios end when all units on one side are eliminated or when the time limit is reached. The objective is to maximize the win rate of the allied agents. Detailed descriptions of the scenarios and unit compositions are provided in Table A.2 and Fig. A.2.

**State and observation spaces**
In the SMAC environment, each agent receives partial observations that contain information about visible allies and enemies within a fixed sight range of 9. These observations do not include any global state and are specifically designed to support decentralized decision-making based only on each agent's local view of the environment.

The global state, which is used during centralized training, is constructed by aggregating the features of all agents and enemies. It consists of three primary components. The ally state includes each agent's relative $x$ and $y$ positions, health, energy, shield (if applicable), and unit type. The enemy state is similar but excludes energy, containing relative positions, health, shield, and unit type of enemies. The last action component records the most recent action taken by each agent, represented as a one-hot encoded vector. The full global state is formed by concatenating these components, and its dimensionality depends on the number of agents, enemies, and available actions.

Each agent's observation vector is separately constructed from the following elements. The movement features indicate the four cardinal directions the agent can move in, resulting in a fixed size of 4. The enemy features describe each observed enemy, including available action flag, distance to the agent, relative $x$ and $y$ positions, health, shield (if applicable), and unit type. The ally features encode the same types of information for all visible allies, excluding the observing agent. Finally, the own features contain the observing agent's own health, shield, and unit type.

The precise dimensions of the observation and state vectors vary depending on the specific map scenario and unit composition, and are summarized in Table A.2.

**Action space**
Agents can perform discrete actions, including movement in four cardinal directions (North, South, East, West), attacking specific enemy units within a shooting range of 6 units, and specialized actions such as healing for units like Medivacs. Additionally, agents can perform a stop or a no-op action, the latter being restricted to dead units.

The size of the action space varies depending on the scenario and is defined as $n_{\text{actions}} = 6 + n_{\text{enemies}}$, where 6 represents movement, stop, or a no-op action. The inclusion of $n_{\text{enemies}}$ accounts for the need to specify which enemy unit is targeted

when performing an attack action. The exact size of the action space varies across different maps, as summarized in Table A.2.

**Reward function**

SMAC uses a shaped reward function $R$ to guide learning, including components for damage dealt ($R_{\text{damage}}$), enemy units killed ($R_{\text{enemy\_killed}}$), and scenario victory ($R_{\text{win}}$). The total reward is defined as:

$$R = \sum_{e \in \text{enemies}} \Delta\text{Health}(e) + \sum_{e \in \text{enemies}} \mathbb{I}(\{\text{Health}(e) = 0\}) \cdot \text{Reward}_{\text{death}} + \mathbb{I}(\{\text{win} = \text{True}\}) \cdot \text{Reward}_{\text{win}}$$

Here, $\text{Health}(e)$ represents the health of an enemy unit $e$, and $\Delta\text{Health}(e)$ is the reduction in its health during a timestep. The indicator function $\mathbb{I}(\cdot)$ returns 1 if the condition inside is true and 0 otherwise. The parameters $\text{Reward}_{\text{death}}$ and $\text{Reward}_{\text{win}}$ are scaling factors for rewards when an enemy unit is killed and when the agents win the scenario, set to 10 and 200, respectively.

| Map | Ally Units | Enemy Units | State Dimension | Obs Dimension | Num. of Actions |
|---|---|---|---|---|---|
| 3m | 3 Marines | 3 Marines | 48 | 30 | 9 |
| 3s_vs_3z | 3 Stalkers | 3 Zealots | 54 | 36 | 9 |
| 2s3z | 2 Stalkers, 3 Zealots | 2 Stalkers, 3 Zealots | 120 | 80 | 11 |
| 8m | 8 Marines | 8 Marines | 168 | 80 | 14 |
| 1c3s5z | 1 Colossus, 3 Stalkers, 5 Zealots | 1 Colossus, 3 Stalkers, 5 Zealots | 270 | 162 | 15 |
| MMM | 1 Medivac, 2 Marauders, 7 Marines | 1 Medivac, 2 Marauders, 7 Marines | 290 | 160 | 16 |

*Table A.2.* The number of agents, the dimensions of state and observation spaces, and the number of actions in SMAC scenarios

# B. Implementation Details

In this section, we provide a detailed implementation of the proposed WALL framework. Section B.1 outlines the implementation of the Transformer used to efficiently identify critical initial steps. Section B.2 elaborates on the components constituting the Wolfpack attack and provides an in-depth explanation of the reinforcement learning implementation.

## B.1. Practical Implementation of Planner Transformer

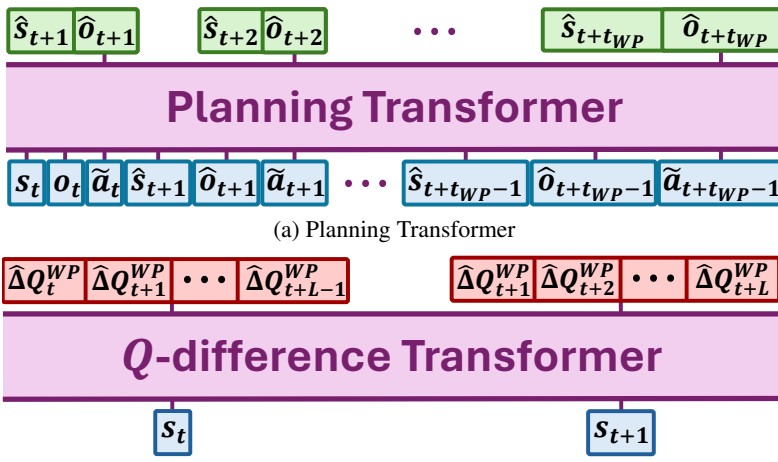

(a) Planning Transformer

(b) $Q$-difference Transformer

*Figure B.1.* Structure of Transformers

Critical attacking step selection introduced in Section 4.4 requires planning to compute the reduction in $Q$-function values, $\Delta Q_l^{\mathrm{WP}}$ ($l = t, \cdots, t+L-1$), over multiple time steps. To facilitate this process, a Transformer model is employed. During training, the Transformer predicts $(\hat{s}_{t+1}, \hat{\mathbf{o}}_{t+1}, \cdots, \hat{s}_{t+t_{\mathrm{WP}}}, \hat{\mathbf{o}}_{t+t_{\mathrm{WP}}})$ at the current step $t$ to compute the target $\hat{\Delta} Q_t^{\mathrm{WP}}$. However, performing this planning process at each evaluation step to calculate the probability of an initial attack, $P_{t,\mathrm{attack}}$, is computationally expensive.

To address this, the single Transformer is split into two components: a planning Transformer and a $Q$-difference Transformer. The planning Transformer predicts $(\hat{s}_{t+1}, \hat{\mathbf{o}}_{t+1})$ and is used only during training, while the $Q$-difference Transformer predicts $\Delta Q_l^{\mathrm{WP}}$ ($l = t, \cdots, t+L-1$) and is employed exclusively during evaluation. This separation enables efficient computation of $P_{t,\mathrm{attack}}$ during evaluation, significantly reducing computational costs. Both Transformers adopt the Decision Transformer structure (Chen et al., 2021), consisting of Transformer decoder layers. The planning Transformer is parameterized by $\phi_{\mathrm{planning}}$, and the $Q$-difference Transformer by $\phi_{\mathrm{qdiff}}$. Their respective loss functions are defined as:

$$\mathcal{L}_{\mathrm{planning}}(\phi_{\mathrm{planning}}) = \mathbb{E}\left[\|s_{t+1} - \hat{s}_{t+1}(\phi_{\mathrm{planning}})\|^2 + \|\mathbf{o}_{t+1} - \hat{\mathbf{o}}_{t+1}(\phi_{\mathrm{planning}})\|^2\right], \tag{B.1}$$

$$\mathcal{L}_{\mathrm{qdiff}}(\phi_{\mathrm{qdiff}}) = \mathbb{E}\left[\|\Delta Q_t^{\mathrm{WP}} - \hat{\Delta} Q_t^{\mathrm{WP}}(\phi_{\mathrm{qdiff}})\|^2\right], \; \forall t. \tag{B.2}$$

As shown in Fig. B.1, the estimated state $\hat{s}_{t+1}$ and observations $\hat{\mathbf{o}}_{t+1}$ are generated by the planning Transformer, which takes previous trajectories as input. Similarly, the estimated $Q$-difference $\hat{\Delta} Q_t^{\mathrm{WP}}$ is produced by the $Q$-diff Transformer, using previous states as input. The loss function $\mathcal{L}_{\mathrm{planning}}$ minimizes the prediction error for the next state $s_{t+1}$ and observation $\mathbf{o}_{t+1}$, while $\mathcal{L}_{\mathrm{qdiff}}$ minimizes the prediction error for $\Delta Q_l^{\mathrm{WP}}$ resulting from the Wolfpack attack. Fig. B.1 illustrates the architectures of the Transformers, where (a) represents the planning Transformer and (b) represents the $Q$-diff Transformer.

## B.2. Detailed Implementation of WALL

The WALL framework trains robust MARL policies to counter the Wolfpack adversarial attack by employing a $Q$-learning approach within the CTDE paradigm. Each agent computes its individual $Q$-values, $Q^i(\tau_t^i, a_t^i)$, $i = 1, \cdots, n$, using separate $Q$-networks. These individual values, along with the global state, are combined through a mixing network to produce the total $Q$-value, $Q_\theta^{tot}$, parameterized by $\theta$. This joint value function ensures effective coordination among agents under adversarial scenarios.

To achieve robustness, the training process minimizes the temporal difference (TD) loss, which incorporates the observed rewards, state transitions, and target $Q$-values parameterized by a separate target network, $\theta^-$. By leveraging this target network, the CTDE frameworks stabilize learning and mitigate the impact of the Wolfpack attack. The TD loss is defined as:

$$\mathcal{L}_{\text{TD}}(\theta) = \mathbb{E}_{s,\mathbf{a},r,s'}\left[\left(r_t + \gamma \max_{\mathbf{a}'} Q_{\theta^-}^{tot}(s_{t+1}, \mathbf{a}') - Q_\theta^{tot}(s_t, \mathbf{a}_t)\right)^2\right], \tag{B.3}$$

where the target network parameter $\theta^-$ is updated by applying the exponential moving average (EMA) to $\theta$. This training mechanism allows agents to adapt and develop robust policies capable of resisting the coordinated disruptions caused by the Wolfpack adversarial attack, ensuring enhanced performance and resilience in MARL scenarios. In addition, we use 3 value-based CTDE algorithms as baselines for the WALL framework: QMIX, VDN, and QPLEX. Below, we provide an outline of the key details of these baseline algorithms:

**Value-Decomposition Networks (VDN)**
VDN (Sunehag et al., 2017) is a $Q$-learning algorithm designed for cooperative MARL. It introduces an additive decomposition of the joint $Q$-value into individual agent $Q$-values, enabling centralized training and decentralized execution. The joint action-value function, $Q^{tot}$, is expressed as:

$$Q^{tot}(s_t, \mathbf{a}_t) = \sum_{i=1}^{n} Q^i(\tau_t^i, a_t^i), \tag{B.4}$$

allowing agents to act independently during execution by relying only on their local $Q^i$ values.

**QMIX**
QMIX (Rashid et al., 2020) extends VDN by introducing a more expressive, non-linear representation of the joint $Q$-value, while maintaining a monotonic relationship between $Q^{tot}$ and individual agent $Q$-values, $Q^i(\tau^i, a^i)$. This ensures individual-global-max (IGM) condition:

$$\frac{\partial Q^{tot}}{\partial Q^i} \geq 0, \forall i, \tag{B.5}$$

guaranteeing consistency between centralized and decentralized policies. Specifically:

$$\arg\max_{\mathbf{a}} Q^{\text{tot}}(s_t, \mathbf{a}_t) = \begin{pmatrix} \arg\max_{a^1} Q^1(\tau_t^1, a_t^1), \\ \vdots \\ \arg\max_{a^n} Q^n(\tau_t^n, a_t^n) \end{pmatrix}. \tag{B.6}$$

QMIX combines agent networks, a mixing network, and hypernetworks, where hypernetworks dynamically parameterize the mixing network based on the global state $s_t$. The weights generated by the hypernetworks are constrained to be non-negative to enforce the monotonicity constraint.

**QPLEX**
QPLEX (Wang et al., 2020b) introduces a duplex dueling architecture to enhance the representation of joint action-value functions while adhering to the IGM principle. QPLEX reformulates the IGM principle in an advantage-based form:

$$\arg\max_{\mathbf{a}} A^{tot}(\tau, \mathbf{a}) = \begin{pmatrix} \arg\max_{a^1} A^1(\tau^1, a^1), \\ \vdots \\ \arg\max_{a^n} A^n(\tau^n, a^n) \end{pmatrix}, \tag{B.7}$$

where $A^{tot}$ and $A^i$ are the advantage functions for joint and individual action-value functions, respectively. The joint action-value function is expressed as:

$$Q^{tot}(\tau, \mathbf{a}) = \sum_{i=1}^{n} Q^i(\tau, a^i) + \sum_{i=1}^{n} (\lambda_i(\tau, \mathbf{a}) - 1) A^i(\tau, a^i). \tag{B.8}$$

where $\lambda_i(\tau, \mathbf{a}) > 0$ are importance weights generated using a multi-head attention mechanism to enhance expressiveness.

Here, VDN and QMIX are implemented using the PyMARL codebase https://github.com/oxwhirl/pymarl, while QPLEX is implemented using its official codebase https://github.com/wjh720/QPLEX.

# C. Experimental Details

All experiments in this paper are conducted on a GPU server equipped with an NVIDIA GeForce RTX 3090 GPU and AMD EPYC 7513 32-Core processors running Ubuntu 20.04 and PyTorch. We follow the implementations and loss scales provided by the CTDE algorithms and focus on parameter searches for hyperparameters related to the proposed Wolfpack adversarial attack. Comparisons are performed using the optimal hyperparameter setup, with an ablation study available in Appendix F.

## C.1. Hyperparameter Setup

We conduct parameter search for the number of Wolfpack attacks $K_{\mathrm{WP}} \in [1, 2, 3, 4]$, the attack duration $t_{\mathrm{WP}} \in [1, 2, 3, 4]$, the number of follow-up agents $m$, and the temperature $T \in [0.1, 0.2, 0.5, 1.0]$. The total number of attacks $K$ is then determined based on $K = K_{\mathrm{WP}} \times (t_{\mathrm{WP}} + 1)$, separated into training and testing setups. During training, $K$ is selected through hyperparameter sweeping to ensure optimal performance. For testing, $K$ is unified across all adversarial attack setups, including Random Attack, EGA, and the Wolfpack Adversarial Attack, to ensure fair comparisons. Additionally, the attack period $L$ is chosen based on the average episode length of SMAC scenarios and the total number of attacks, with $L = 20$ is fixed as appropriate. Transformer hyperparameters, shared between the Planning Transformer and $Q$-difference Transformer, such as the number of heads, decoder layers, embedding dimensions, and input sequence length, are selected to balance accuracy and computational efficiency.

The $Q$-learning hyperparameters (shared across all CTDE methods) and those specific to the CTDE algorithms are detailed in Table C.1 and Table C.2, respectively. The Wolfpack adversarial attack-related hyperparameters for the WALL framework, shared across all SMAC scenarios and scenario-specific setups, are presented in Table C.3.

| Hyperparameter | Value |
|---|---|
| Epsilon | $1.0 \rightarrow 0.05$ |
| Epsilon Anneal Time | 50000 timesteps |
| Train Interval | 1 episode |
| Gamma | 0.99 |
| Critic Loss | MSE Loss |
| Buffer Size | 5000 episodes |
| Batch Size | 32 episodes |
| Agent Learning Rate | 0.0005 |
| Critic Learning Rate | 0.0005 |
| Optimizer | RMSProp |
| Optimizer Alpha | 0.99 |
| Optimizer Eps | 1e-5 |
| Gradient Clip Norm | 10.0 |
| Num GRU Layers | 1 |
| RNN Hidden State Dim | 64 |
| Double Q | True |

*Table C.1.* Common $Q$-learning hyperparameters

| Hyperparameter | VDN | QMIX | QPLEX |
|---|---|---|---|
| Mixer | VDN | QMIX | QPLEX |
| Mixing Embed Dim. | - | 32 | 32 |
| Hypernet Layers | - | 2 | 2 |
| Hypernet Embed Dim. | - | 64 | 64 |
| Adv Hypernet Layers | - | - | 1 |
| Adv Hypernet Embed Dim. | - | - | 64 |
| Num. Kernel | - | - | 2 |

*Table C.2.* VDN, QMIX, QPLEX hyperparameters

| Common Hyperparameters | Value |
|---|---|
| Attack duration ($t_{\mathrm{WP}}$) | 3 |
| Temperature ($T$) | 0.5 |
| Attack Period ($L$) | 20 |
| Num. Transformer Head | 1 |
| Num. Transformer Decoder Layer | 1 |
| Transformer Embed Dim. | 64 |
| Input Sequence Length | 20 |

| Scenario | PP_3/1 | PP_6/2 | PP_9/3 |
|---|---|---|---|
| Num. Total Attacks (Train) ($K$) | 4 | 4 | 4 |
| Num. Total Attacks (Test) ($K$) | 4 | 4 | 4 |
| Num. Wolfpack Attacks ($K_{\mathrm{WP}}$) | 1 | 1 | 1 |
| Num. Follow-up Agents ($m$) | 1 | 3 | 5 |

| Scenario | 3m | 3s_vs_3z | 2s3z | 8m | 1c3s5z | MMM |
|---|---|---|---|---|---|---|
| Num. Total Attacks (Train) ($K$) | 8 | 16 | 12 | 8 | 16 | 16 |
| Num. Total Attacks (Test) ($K$) | 8 | 4 | 8 | 4 | 8 | 8 |
| Num. Wolfpack Attacks ($K_{\mathrm{WP}}$) | 2 | 4 | 3 | 2 | 4 | 4 |
| Num. Follow-up Agents ($m$) | 1 | 1 | 2 | 3 | 4 | 4 |

*Table C.3.* Wolfpack hyperparameters shared across scenarios and scenario-specific values

# D. Details of Other Robust MARL Methods

In this section, we detail various robust MARL methods compared against the proposed WALL framework, as below:

**Robust Adversarial Reinforcement Learning (RARL)**
RARL (Pinto et al., 2017) enhances policy robustness by training a protagonist agent and an adversary in a two-player zero-sum Markov game. At each timestep $t$, the agents observe state $s_t$ and take actions $a_t^1 \sim \mu(s_t)$ and $a_t^2 \sim \nu(s_t)$, where $\mu$ is the protagonist's policy, and $\nu$ is the adversary's policy. The state transitions follow:

$$s_{t+1} = P(s_t, a_t^1, a_t^2). \tag{D.9}$$

The protagonist maximizes its cumulative reward $R^1$, while the adversary minimizes it:

$$R_*^1 = \min_\nu \max_\mu R^1(\mu, \nu) = \max_\mu \min_\nu R^1(\mu, \nu). \tag{D.10}$$

**Robustness via Adversary Populations (RAP)**
RAP (Vinitsky et al., 2020) improves robustness by training agents against a population of adversaries, reducing overfitting to specific attack patterns. During training, an adversary is sampled uniformly from the population $\pi_{\phi_1}, \pi_{\phi_2}, \ldots, \pi_{\phi_n}$. The objective is:

$$\max_\theta \min_{\phi_1, \ldots, \phi_n} \mathbb{E}_{i \sim U(1,n)} \left[ \sum_{t=0}^T \gamma^t r(s_t, a_t, \alpha a_t^i) \middle| \pi_\theta, \pi_{\phi_i} \right], \tag{D.11}$$

where $\pi_\theta$ is the agent's policy, $\pi_{\phi_i}$ is the $i$-th adversary, and $\alpha$ controls adversary strength.

**Robust Multi-Agent Coordination via Evolutionary Generation of Auxiliary Adversarial Attackers (ROMANCE)**
ROMANCE (Yuan et al., 2023) generates diverse auxiliary adversarial attackers to improve robustness in CMARL. Its objective combines attack quality and diversity:

$$L_{\text{adv}}(\phi) = \frac{1}{n_p} \sum_{j=1}^{n_p} L_{\text{opt}}(\phi_j) - \alpha L_{\text{div}}(\phi), \tag{D.12}$$

where $L_{\text{opt}}$ minimizes the ego-system's return, $L_{\text{div}}$ promotes diversity using Jensen-Shannon Divergence, and $n_p$ is the number of adversarial policies. ROMANCE uses an evolutionary mechanism to explore diverse attacks.

We implement RARL and RAP for multi-agent systems, as well as ROMANCE with EGA, using the ROMANCE codebase available at https://github.com/zzq-bot/ROMANCE.

**ERNIE**
ERNIE (Bukharin et al., 2024) improves robustness by promoting Lipschitz continuity through adversarial regularization. It minimizes discrepancies between policy outputs under perturbed and non-perturbed observations:

$$R_\pi(o_k; \theta_k) = \max_{\|\delta\| \leq \epsilon} D(\pi_{\theta_k}(o_k + \delta), \pi_{\theta_k}(o_k)), \tag{D.13}$$

where $o_k$ is the agent's observation, $\delta$ is a bounded perturbation, and $D$ measures divergence (e.g., KL-divergence). ERNIE reformulates adversarial training as a Stackelberg game and extends its framework to mean-field MARL for scalability in large-agent settings. We evaluate ERNIE using its official codebase at https://github.com/abukharin3/ERNIE.

# E. Additional Experiments Results

This section presents additional experimental results. E.1 reports comparison results for other CTDE algorithms, E.2 provides learning curves across additional SMAC scenarios, E.3 presents a performance comparison for EGA, E.4 discusses computational cost, and E.5 includes general robustness experiments.

## E.1. Comparison Results for Other CTDE Algorithms

Our proposed Wolfpack attack is compatible with various value-based MARL algorithms. Experimental results in this section demonstrate its significant impact on robustness, not only in QMIX, as discussed in the main text, but also in VDN and QPLEX. The hyperparameters used in these experiments follow those outlined in Appendix C.1, with WALL hyperparameters remaining consistent across all algorithms, including QMIX.

**VDN Results:** Table E.1 shows the average win rates for various robust MARL methods with VDN against attacker baselines. Both models and attackers are trained using the VDN-based algorithm. Results indicate that the proposed Wolfpack attack is more detrimental than Random Attack or EGA for VDN. For example, while EGA reduces Vanilla VDN's performance by $95.6\% - 73.8\% = 21.8\%$, the Wolfpack attack causes a larger reduction of $95.6\% - 44.1\% = 51.5\%$, demonstrating its severity. Additionally, the WALL framework outperforms other baselines under both Natural conditions and adversarial attacks, showcasing its robustness.

| Method | Scenario | 2s3z | 3m | 3s_vs_3z | 8m | MMM | 1c3s5z | Mean |
|---|---|---|---|---|---|---|---|---|
| Natural | Vanilla VDN | $98.5 \pm 1.4$ | $96.0 \pm 3.0$ | $99.3 \pm 0.5$ | $97.8 \pm 2.0$ | $98.3 \pm 0.5$ | $83.5 \pm 9.5$ | $95.6 \pm 1.7$ |
| | RANDOM | $99.0 \pm 0.1$ | $\mathbf{99.8 \pm 0.1}$ | $99.4 \pm 0.5$ | $\mathbf{98.9 \pm 0.1}$ | $98.8 \pm 1.0$ | $97.6 \pm 0.5$ | $98.9 \pm 0.4$ |
| | RARL | $95.3 \pm 0.5$ | $92.5 \pm 3.4$ | $99.3 \pm 0.5$ | $96.3 \pm 1.5$ | $93.2 \pm 1.5$ | $91.8 \pm 0.3$ | $94.7 \pm 0.6$ |
| | RAP | $93.4 \pm 1.3$ | $96.8 \pm 1.8$ | $99.3 \pm 0.5$ | $98.7 \pm 1.0$ | $97.3 \pm 0.5$ | $97.3 \pm 0.5$ | $97.1 \pm 0.4$ |
| | ERNIE | $94.6 \pm 3.9$ | $99.4 \pm 0.5$ | $97.1 \pm 0.5$ | $98.4 \pm 1.4$ | $98.8 \pm 0.8$ | $96.5 \pm 1.7$ | $97.5 \pm 0.9$ |
| | ROMANCE | $98.2 \pm 0.1$ | $97.9 \pm 1.1$ | $99.4 \pm 0.5$ | $93.5 \pm 5.4$ | $97.9 \pm 1.0$ | $93.0 \pm 3.1$ | $96.6 \pm 0.8$ |
| | WALL (ours) | $\mathbf{99.8 \pm 0.1}$ | $99.4 \pm 0.5$ | $\mathbf{99.9 \pm 0.1}$ | $96.9 \pm 2.0$ | $\mathbf{99.4 \pm 0.5}$ | $\mathbf{100.0 \pm 0.1}$ | $\mathbf{99.2 \pm 0.2}$ |
| Random Attack | Vanilla VDN | $80.5 \pm 1.5$ | $65.5 \pm 6.5$ | $96.5 \pm 0.5$ | $52.5 \pm 10.5$ | $95.0 \pm 1.0$ | $76.5 \pm 7.5$ | $77.8 \pm 1.9$ |
| | RANDOM | $83.0 \pm 1.0$ | $\mathbf{94.5 \pm 1.5}$ | $96.0 \pm 0.7$ | $88.5 \pm 0.5$ | $95.5 \pm 2.5$ | $93.5 \pm 1.5$ | $91.8 \pm 0.5$ |
| | RARL | $81.0 \pm 4.2$ | $64.5 \pm 8.4$ | $97.3 \pm 0.5$ | $72.9 \pm 0.1$ | $76.3 \pm 7.5$ | $92.0 \pm 0.2$ | $80.7 \pm 2.1$ |
| | RAP | $89.8 \pm 3.1$ | $75.8 \pm 37.9$ | $97.7 \pm 1.9$ | $81.2 \pm 0.5$ | $95.2 \pm 2.6$ | $92.3 \pm 0.5$ | $88.7 \pm 1.6$ |
| | ERNIE | $80.2 \pm 2.0$ | $66.9 \pm 2.8$ | $93.1 \pm 1.7$ | $67.1 \pm 4.9$ | $89.1 \pm 2.1$ | $90.4 \pm 4.5$ | $81.1 \pm 1.2$ |
| | ROMANCE | $91.0 \pm 5.0$ | $79.0 \pm 6.0$ | $98.4 \pm 0.4$ | $54.0 \pm 5.0$ | $97.5 \pm 0.5$ | $92.0 \pm 3.0$ | $85.3 \pm 0.6$ |
| | WALL (ours) | $\mathbf{95.5 \pm 1.5}$ | $86.0 \pm 6.0$ | $\mathbf{99.4 \pm 0.4}$ | $\mathbf{90.0 \pm 3.0}$ | $\mathbf{98.5 \pm 0.5}$ | $\mathbf{98.1 \pm 0.1}$ | $\mathbf{94.6 \pm 1.4}$ |
| EGA | Vanilla VDN | $69.5 \pm 7.5$ | $54.5 \pm 12.5$ | $94.5 \pm 1.5$ | $59.5 \pm 12.5$ | $90.5 \pm 2.5$ | $74.5 \pm 9.5$ | $73.8 \pm 1.0$ |
| | RANDOM | $56.0 \pm 6.0$ | $73.0 \pm 6.0$ | $84.0 \pm 13.0$ | $84.5 \pm 6.5$ | $85.5 \pm 1.5$ | $86.5 \pm 3.5$ | $78.3 \pm 2.4$ |
| | RARL | $58.0 \pm 1.2$ | $79.0 \pm 2.9$ | $96.5 \pm 0.7$ | $76.7 \pm 4.0$ | $75.9 \pm 8.2$ | $81.3 \pm 4.5$ | $77.9 \pm 1.9$ |
| | RAP | $79.9 \pm 3.1$ | $93.0 \pm 4.6$ | $97.3 \pm 0.5$ | $85.7 \pm 1.3$ | $91.8 \pm 3.8$ | $87.2 \pm 1.5$ | $89.3 \pm 0.2$ |
| | ERNIE | $62.0 \pm 14.9$ | $62.5 \pm 8.2$ | $89.1 \pm 6.6$ | $74.7 \pm 1.6$ | $89.3 \pm 2.5$ | $82.2 \pm 1.9$ | $76.6 \pm 3.6$ |
| | ROMANCE | $86.5 \pm 2.5$ | $\mathbf{93.8 \pm 3.0}$ | $98.0 \pm 0.2$ | $76.5 \pm 0.5$ | $95.5 \pm 2.5$ | $92.0 \pm 0.1$ | $90.3 \pm 0.6$ |
| | WALL (ours) | $\mathbf{91.5 \pm 0.5}$ | $91.0 \pm 2.0$ | $\mathbf{99.0 \pm 1.0}$ | $\mathbf{90.0 \pm 4.0}$ | $\mathbf{97.5 \pm 0.5}$ | $\mathbf{95.5 \pm 0.5}$ | $\mathbf{94.1 \pm 0.1}$ |
| Wolfpack Adversarial Attack (ours) | Vanilla VDN | $54.0 \pm 4.0$ | $20.5 \pm 5.5$ | $91.5 \pm 0.5$ | $24.5 \pm 12.5$ | $18.5 \pm 9.5$ | $55.5 \pm 2.5$ | $44.1 \pm 1.1$ |
| | RANDOM | $47.0 \pm 4.0$ | $89.0 \pm 1.0$ | $90.0 \pm 5.0$ | $41.0 \pm 14.0$ | $18.5 \pm 5.5$ | $83.5 \pm 0.5$ | $61.5 \pm 2.7$ |
| | RARL | $59.5 \pm 8.7$ | $41.3 \pm 16.4$ | $96.8 \pm 0.9$ | $13.1 \pm 2.0$ | $24.3 \pm 6.5$ | $61.6 \pm 11.7$ | $49.4 \pm 1.8$ |
| | RAP | $64.3 \pm 3.5$ | $67.7 \pm 33.8$ | $98.9 \pm 0.1$ | $23.9 \pm 6.1$ | $53.3 \pm 2.5$ | $82.1 \pm 5.4$ | $65.0 \pm 1.7$ |
| | ERNIE | $33.1 \pm 3.6$ | $25.8 \pm 6.5$ | $93.0 \pm 4.5$ | $17.2 \pm 9.5$ | $23.5 \pm 9.3$ | $66.4 \pm 13.5$ | $43.2 \pm 1.7$ |
| | ROMANCE | $63.0 \pm 10.0$ | $46.5 \pm 20.5$ | $97.5 \pm 0.5$ | $19.5 \pm 7.5$ | $38.0 \pm 5.0$ | $83.0 \pm 4.0$ | $57.9 \pm 3.1$ |
| | WALL (ours) | $\mathbf{91.5 \pm 2.5}$ | $\mathbf{91.5 \pm 2.5}$ | $\mathbf{100.0 \pm 0.0}$ | $\mathbf{71.5 \pm 1.5}$ | $\mathbf{98.0 \pm 1.0}$ | $\mathbf{93.5 \pm 4.5}$ | $\mathbf{91.0 \pm 0.8}$ |

*Table E.1.* Average test win rates of robust MARL policies under various attack settings (VDN)

**QPLEX Results:** Similarly, Table E.2 reports the average win rates various robust MARL methods with QPLEX against attacker baselines. Both models and attackers are trained using the QPLEX-based algorithm. Results reveal that the Wolfpack attack is also more detrimental for QPLEX compared to Random Attack and EGA. For instance, EGA reduces Vanilla QPLEX's performance by $98.4\% - 57.2\% = 41.2\%$, whereas the Wolfpack attack results in a larger reduction of $98.4\% - 33.1\% = 65.3\%$. The WALL framework again demonstrates superior robustness, performing well against all attacks, including the Wolfpack attack.

| Method | Scenario | 2s3z | 3m | 3s_vs_3z | 8m | MMM | 1c3s5z | Mean |
|---|---|---|---|---|---|---|---|---|
| Natural | Vanilla QPLEX | $97.1 \pm 1.3$ | $\mathbf{99.2 \pm 0.5}$ | $99.4 \pm 0.5$ | $97.4 \pm 0.3$ | $99.5 \pm 0.5$ | $97.1 \pm 0.9$ | $98.4 \pm 0.1$ |
| | RANDOM | $97.7 \pm 2.1$ | $98.7 \pm 0.8$ | $99.6 \pm 0.1$ | $99.1 \pm 0.9$ | $98.2 \pm 1.2$ | $98.5 \pm 1.3$ | $98.7 \pm 0.5$ |
| | RARL | $94.4 \pm 4.5$ | $89.7 \pm 2.0$ | $88.4 \pm 6.8$ | $97.8 \pm 1.6$ | $98.8 \pm 0.8$ | $94.6 \pm 1.6$ | $94.0 \pm 1.4$ |
| | RAP | $96.5 \pm 1.7$ | $96.6 \pm 2.5$ | $93.2 \pm 0.9$ | $99.1 \pm 0.5$ | $98.8 \pm 0.8$ | $97.8 \pm 0.8$ | $97.7 \pm 0.4$ |
| | ERNIE | $96.7 \pm 2.5$ | $98.8 \pm 0.8$ | $99.7 \pm 0.1$ | $98.8 \pm 1.4$ | $99.1 \pm 0.5$ | $98.4 \pm 1.0$ | $98.6 \pm 0.6$ |
| | ROMANCE | $97.1 \pm 2.9$ | $93.2 \pm 5.9$ | $98.8 \pm 0.8$ | $94.7 \pm 3.2$ | $99.0 \pm 1.1$ | $99.0 \pm 1.1$ | $96.8 \pm 1.2$ |
| | WALL (ours) | $\mathbf{99.5 \pm 0.5}$ | $97.7 \pm 2.1$ | $\mathbf{99.9 \pm 0.1}$ | $\mathbf{99.8 \pm 0.1}$ | $99.0 \pm 0.7$ | $\mathbf{99.5 \pm 0.6}$ | $\mathbf{99.2 \pm 0.3}$ |
| Random Attack | Vanilla QPLEX | $75.2 \pm 2.5$ | $36.2 \pm 6.6$ | $81.9 \pm 16.0$ | $40.0 \pm 9.8$ | $65.3 \pm 6.2$ | $75.0 \pm 2.8$ | $62.3 \pm 5.6$ |
| | RANDOM | $85.9 \pm 0.9$ | $69.0 \pm 5.8$ | $96.2 \pm 1.6$ | $89.0 \pm 7.7$ | $91.2 \pm 2.4$ | $95.5 \pm 1.2$ | $87.8 \pm 0.7$ |
| | RARL | $81.2 \pm 13.6$ | $61.3 \pm 13.9$ | $74.7 \pm 12.6$ | $73.7 \pm 21.6$ | $92.0 \pm 0.8$ | $92.0 \pm 2.9$ | $79.1 \pm 4.3$ |
| | RAP | $90.8 \pm 2.8$ | $78.0 \pm 7.0$ | $92.1 \pm 10.3$ | $65.3 \pm 4.5$ | $95.0 \pm 1.4$ | $95.7 \pm 1.9$ | $85.0 \pm 1.9$ |
| | ERNIE | $82.0 \pm 2.0$ | $59.0 \pm 9.0$ | $93.0 \pm 1.6$ | $80.0 \pm 1.4$ | $91.3 \pm 0.9$ | $93.0 \pm 0.8$ | $83.1 \pm 1.9$ |
| | ROMANCE | $89.2 \pm 2.1$ | $64.9 \pm 13.4$ | $93.7 \pm 2.1$ | $51.6 \pm 5.3$ | $92.2 \pm 4.1$ | $95.2 \pm 1.0$ | $76.0 \pm 2.8$ |
| | WALL (ours) | $\mathbf{97.4 \pm 0.9}$ | $\mathbf{85.4 \pm 2.5}$ | $\mathbf{99.4 \pm 0.5}$ | $\mathbf{92.9 \pm 3.5}$ | $\mathbf{98.3 \pm 1.2}$ | $\mathbf{98.6 \pm 1.2}$ | $\mathbf{95.3 \pm 0.7}$ |
| EGA | Vanilla QPLEX | $48.1 \pm 3.4$ | $16.0 \pm 5.1$ | $72.5 \pm 15.1$ | $58.5 \pm 16.8$ | $71.0 \pm 4.2$ | $76.9 \pm 1.5$ | $57.2 \pm 6.2$ |
| | RANDOM | $60.5 \pm 6.8$ | $61.4 \pm 14.9$ | $82.0 \pm 6.6$ | $86.2 \pm 3.4$ | $85.3 \pm 2.5$ | $88.8 \pm 2.5$ | $77.4 \pm 3.1$ |
| | RARL | $63.4 \pm 0.5$ | $65.7 \pm 6.2$ | $71.0 \pm 8.2$ | $86.0 \pm 6.5$ | $89.7 \pm 1.9$ | $84.7 \pm 2.1$ | $76.7 \pm 1.0$ |
| | RAP | $78.5 \pm 4.0$ | $83.3 \pm 0.9$ | $85.4 \pm 5.3$ | $89.0 \pm 2.2$ | $92.7 \pm 1.7$ | $96.7 \pm 0.5$ | $88.0 \pm 0.7$ |
| | ERNIE | $64.4 \pm 7.0$ | $67.3 \pm 10.3$ | $51.7 \pm 9.5$ | $86.7 \pm 4.7$ | $86.0 \pm 5.0$ | $89.3 \pm 4.5$ | $74.2 \pm 3.0$ |
| | ROMANCE | $79.5 \pm 6.0$ | $80.3 \pm 1.7$ | $90.3 \pm 1.3$ | $80.7 \pm 8.3$ | $95.2 \pm 2.0$ | $92.5 \pm 1.3$ | $85.6 \pm 1.0$ |
| | WALL (ours) | $\mathbf{89.0 \pm 3.8}$ | $\mathbf{83.9 \pm 2.9}$ | $\mathbf{99.6 \pm 0.6}$ | $\mathbf{94.4 \pm 1.2}$ | $\mathbf{96.4 \pm 1.2}$ | $\mathbf{96.1 \pm 0.7}$ | $\mathbf{93.2 \pm 0.7}$ |
| Wolfpack Adversarial Attack (ours) | Vanilla QPLEX | $30.8 \pm 4.3$ | $11.8 \pm 7.5$ | $63.7 \pm 24.3$ | $30.7 \pm 11.1$ | $20.0 \pm 12.5$ | $41.9 \pm 6.1$ | $33.1 \pm 7.1$ |
| | RANDOM | $50.5 \pm 2.9$ | $16.8 \pm 6.8$ | $89.5 \pm 5.7$ | $45.6 \pm 18.0$ | $34.0 \pm 14.1$ | $82.5 \pm 11.3$ | $53.2 \pm 4.9$ |
| | RARL | $55.5 \pm 2.1$ | $27.3 \pm 9.2$ | $78.0 \pm 4.3$ | $46.0 \pm 14.9$ | $21.3 \pm 14.4$ | $79.7 \pm 6.9$ | $51.3 \pm 2.8$ |
| | RAP | $59.0 \pm 4.7$ | $50.3 \pm 14.7$ | $86.7 \pm 8.2$ | $33.7 \pm 3.1$ | $45.0 \pm 7.0$ | $93.7 \pm 2.4$ | $56.3 \pm 3.2$ |
| | ERNIE | $52.9 \pm 6.3$ | $49.0 \pm 9.9$ | $76.0 \pm 9.9$ | $42.7 \pm 10.6$ | $20.7 \pm 10.4$ | $78.7 \pm 6.7$ | $53.3 \pm 3.6$ |
| | ROMANCE | $57.3 \pm 8.4$ | $38.5 \pm 13.9$ | $89.7 \pm 1.3$ | $28.6 \pm 1.1$ | $46.2 \pm 10.6$ | $83.5 \pm 5.5$ | $50.8 \pm 0.9$ |
| | WALL (ours) | $\mathbf{88.3 \pm 1.2}$ | $\mathbf{87.6 \pm 4.7}$ | $\mathbf{99.7 \pm 0.5}$ | $\mathbf{84.5 \pm 1.7}$ | $\mathbf{96.3 \pm 3.9}$ | $\mathbf{99.3 \pm 0.9}$ | $\mathbf{92.6 \pm 1.3}$ |

*Table E.2.* Average test win rates of robust MARL policies under various attack settings (QPLEX)

**Learning Curves for VDN and QPLEX:** We also analyze training curves and average test win rates across different CTDE algorithms. Graphs for the `8m` and `MMM` environments illustrate the average win rates of each policy over training steps under unseen Wolfpack adversarial attacks. Fig. E.1 presents training curves for VDN, while Fig. E.2 shows results for QPLEX. These curves highlight that WALL not only achieves greater robustness but also adapts more quickly to attacks across VDN and QPLEX, further confirming its effectiveness beyond QMIX.

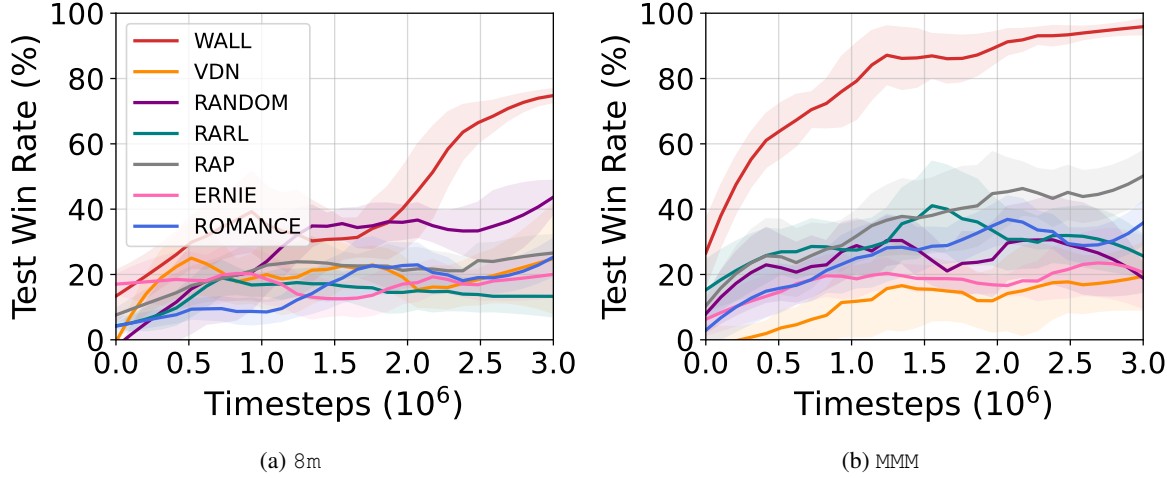

(a) `8m`                                    (b) `MMM`

*Figure E.1.* Learning curves of MARL methods for Wolfpack attack (VDN)

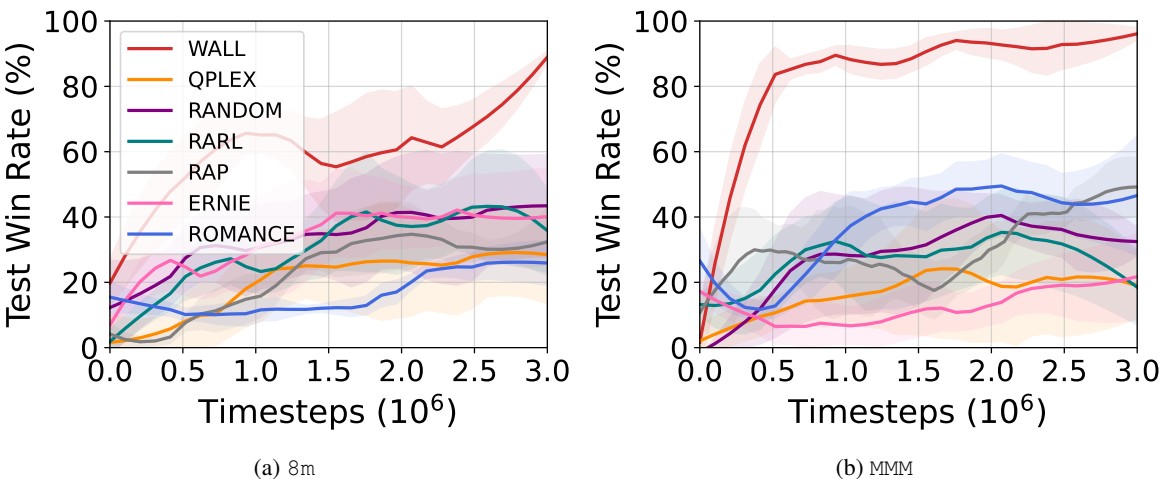

(a) `8m`                                    (b) `MMM`

*Figure E.2.* Learning curves of MARL methods for Wolfpack attack (QPLEX)

### E.2. Learning Curves Across Additional SMAC Scenarios

In this section, we provide the training performance of WALL under Wolfpack adversarial attack across additional SMAC scenarios beyond the `8m` and `MMM` environments, which are emphasized in the main text for their significant performance differences. Fig. E.3 illustrates the training curves for 6 scenarios: `3m`, `3s_vs_3z`, `2s3z`, `8m`, `MMM`, and `1c3s5z`. The results demonstrate that WALL consistently outperforms baseline methods, achieving superior win rates across all scenarios. Additionally, policies trained with WALL adapt more quickly to the challenges posed by Wolfpack attack, showing robust and efficient performance across environments of varying complexity. These findings highlight the effectiveness of WALL in enhancing the robustness of MARL policies against coordinated adversarial attacks.

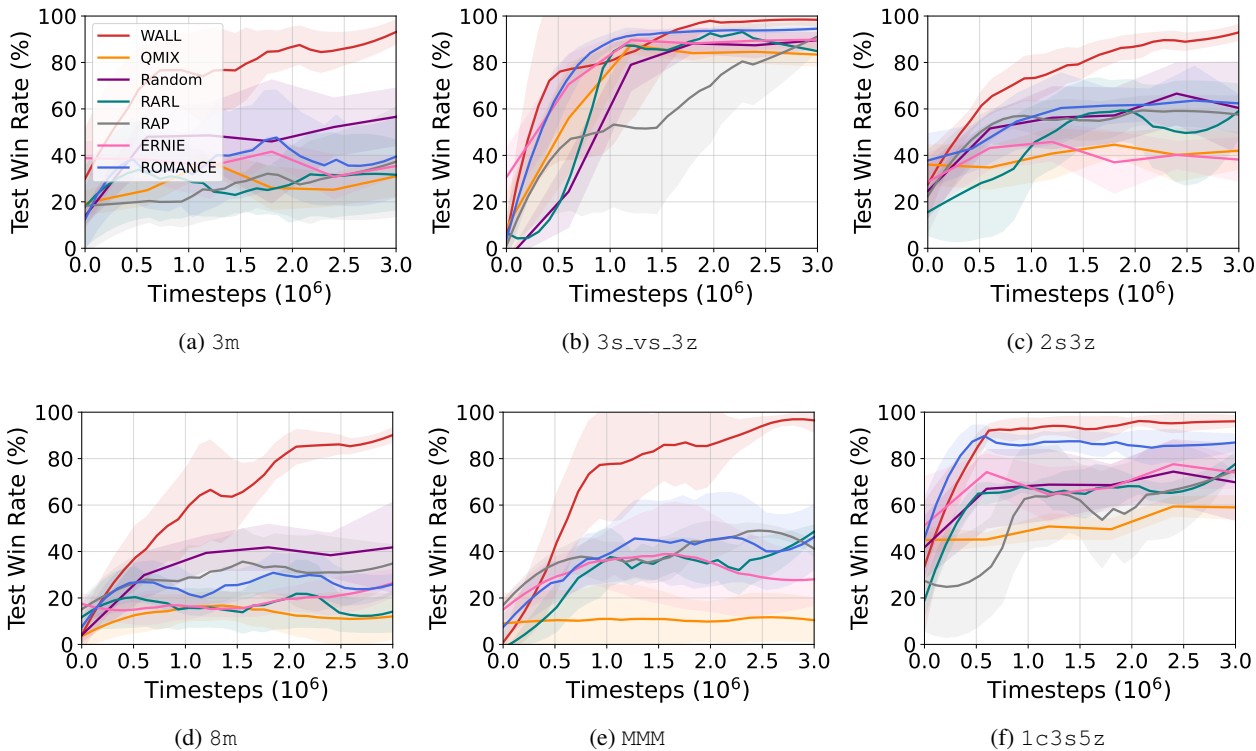

*Figure E.3.* Learning curves of MARL methods for Wolfpack attack across 6 SMAC scenarios (QMIX)

### E.3. Performance Comparison for EGA

This section presents the training performance of WALL under the existing Evolutionary Generation-based Attackers (EGA) (Yuan et al., 2023) method across six SMAC scenarios, as shown in Fig. E.4. The results demonstrate that WALL consistently achieves superior robustness across all scenarios, even against unseen EGA adversaries that are not included in its training process.

The EGA framework generates a diverse and high-quality population of adversarial attackers. Unlike single-adversary methods, EGA maintains an evolving archive of attackers optimized for both quality and diversity, ensuring robust evaluations against various attack strategies. During training, attackers are randomly selected from the archive to simulate diverse attack scenarios. The archive is iteratively updated by replacing low-quality or redundant attackers with newly generated ones. For evaluation, an attacker policy is randomly selected from the archive. The chosen attacker identifies critical attack steps and targets specific victim agents, introducing action perturbations to reduce their individual $Q$-values. WALL demonstrates strong resilience in these challenging environments, effectively mitigating the impact of EGA adversaries and maintaining high performance across all evaluated scenarios.

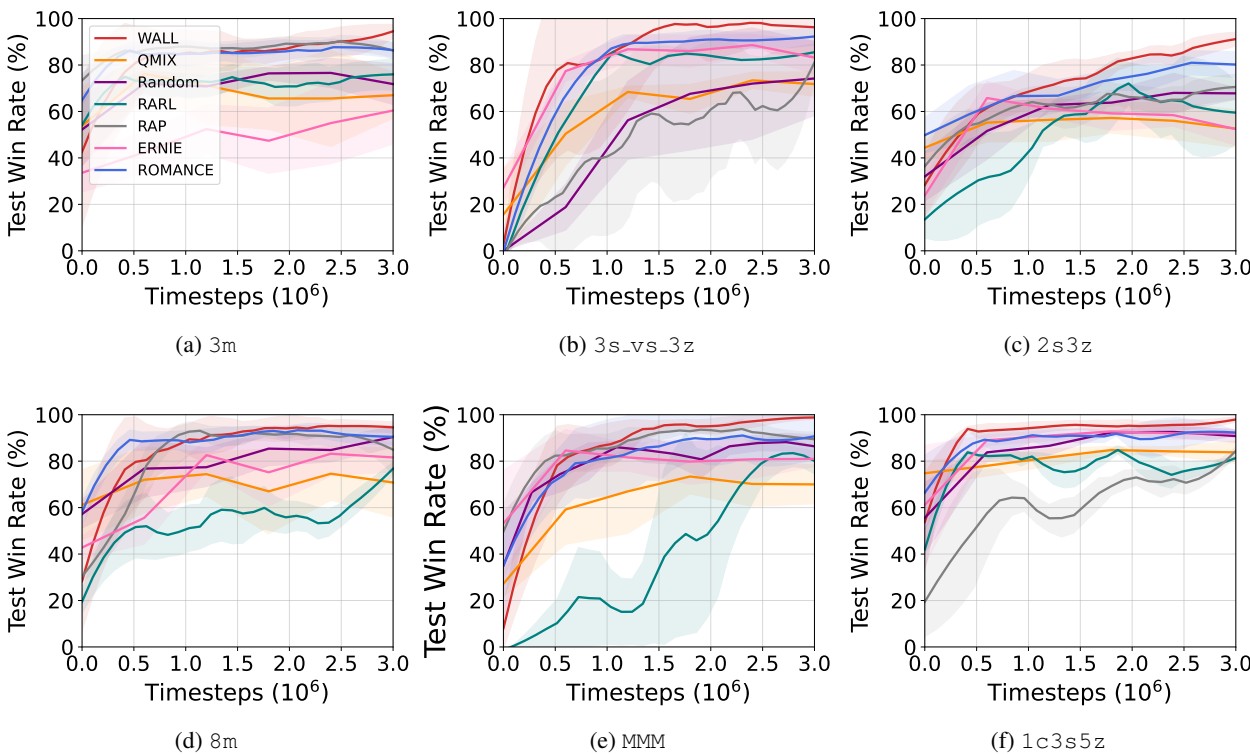

*Figure E.4.* Learning curves of MARL methods for EGA across 6 SMAC scenarios (QMIX)

### E.4. Computational Cost

We measured the total training time (for 3M steps) required for each algorithm in the SMAC environments. The results are summarized in Table E.3. WALL requires approximately 30% more time than ROMANCE, primarily due to the use of the Transformer. However, despite this additional cost, the Transformer-based critical step selection plays an essential role in the effectiveness of our method, as it enables precise identification of attack timing where coordinated perturbations are most disruptive. This capability allows the robust policy to anticipate and defend against strategically timed adversarial threats, ultimately resulting in stronger robustness. In general, training on the MMM map takes longer than on the 8m map due to the increased number of agents, which leads to higher computational costs as a result of scalability.

| Method | Scenario | 8m | MMM |
|---|---|---|---|
| Training Time | Vanilla QMIX | 6h 30m | 7h 30m |
| | RANDOM | 6h 30m | 7h 35m |
| | RARL | 11h 10m | 15h 50m |
| | RAP | 14h 35m | 17h 25m |
| | ERNIE | 16h 30m | 20h 55m |
| | ROMANCE | 16h 35m | 18h 45m |
| | WALL (ours) | 21h 05m | 23h 30m |

*Table E.3.* Training time comparison on 8m and MMM

### E.5. General Robustness Experiments

We conducted additional experiments to evaluate robustness under other commonly used criteria in the 8m and MMM environments. Specifically, we considered: (1) Gaussian observation noise: Standard Gaussian noise (mean 0, standard deviation 1) is injected into agents' local observations, where attack steps (total attack steps : 8 for 8m, 16 for MMM) and the attack group are selected randomly; and (2) Different parameterization in the SMAC test environment: Robustness is evaluated under perturbed unit attributes by reducing the initial health of allied units by 10%, 15%, and 20% in the test setup, compared to the training configuration. These types of noise and perturbations are standard in robustness evaluations and enable us to assess how well the proposed method generalizes beyond action-level perturbations. The Table E.4 below summarizes the results. In both settings, WALL demonstrates consistently stronger performance than existing baselines, suggesting that it generalizes well to broader forms of distributional shift.

| Method | Scenario | 8m | MMM | Mean |
|---|---|---|---|---|
| Gaussian Obs. Noise | Vanilla QMIX | $62.6 \pm 2.3$ | $75.3 \pm 3.1$ | $69.0 \pm 2.1$ |
| | RANDOM | $72.3 \pm 4.2$ | $79.3 \pm 3.9$ | $75.8 \pm 7.2$ |
| | ROMANCE | $69.6 \pm 11.2$ | $76.6 \pm 7.5$ | $73.1 \pm 6.6$ |
| | WALL (ours) | $\mathbf{91.3 \pm 3.3}$ | $\mathbf{97.3 \pm 1.7}$ | $\mathbf{94.3 \pm 3.3}$ |
| Ally HP ↓ 10% | Vanilla QMIX | $52.8 \pm 4.8$ | $88.4 \pm 2.7$ | $70.6 \pm 1.8$ |
| | RANDOM | $50.8 \pm 8.2$ | $95.4 \pm 2.5$ | $73.1 \pm 5.2$ |
| | ROMANCE | $56.8 \pm 16.0$ | $92.4 \pm 4.5$ | $74.6 \pm 8.5$ |
| | WALL (ours) | $\mathbf{73.2 \pm 7.5}$ | $\mathbf{98.6 \pm 1.1}$ | $\mathbf{85.9 \pm 3.7}$ |
| Ally HP ↓ 15% | Vanilla QMIX | $47.4 \pm 6.9$ | $69.0 \pm 5.4$ | $58.2 \pm 5.8$ |
| | RANDOM | $49.2 \pm 4.5$ | $81.4 \pm 5.5$ | $65.3 \pm 4.6$ |
| | ROMANCE | $57.6 \pm 15.7$ | $81.2 \pm 3.5$ | $69.4 \pm 7.8$ |
| | WALL (ours) | $\mathbf{69.2 \pm 10.5}$ | $\mathbf{94.0 \pm 3.2}$ | $\mathbf{81.6 \pm 5.7}$ |
| Ally HP ↓ 20% | Vanilla QMIX | $0.3 \pm 0.4$ | $41.2 \pm 4.5$ | $20.8 \pm 2.1$ |
| | RANDOM | $0.0 \pm 0.0$ | $65.0 \pm 2.7$ | $32.5 \pm 1.3$ |
| | ROMANCE | $2.3 \pm 0.4$ | $57.0 \pm 8.9$ | $29.7 \pm 4.3$ |
| | WALL (ours) | $\mathbf{4.3 \pm 1.9}$ | $\mathbf{89.6 \pm 5.6}$ | $\mathbf{47.0 \pm 3.3}$ |

*Table E.4.* Average test win rates of robust MARL policies under various perturbation settings

## F. Additional Ablation studies

In this section, we provide additional ablation studies on the number of Wolfpack adversarial attacks $K_{\mathrm{WP}}$ and the attack duration $t_{\mathrm{WP}}$ in the 8m and MMM environments, where the performance differences between WALL and other robust MARL methods are most pronounced.

**Number of Wolfpack Attacks** $K_{\mathrm{WP}}$**:** The hyperparameter $K_{\mathrm{WP}}$ determines the number of Wolfpack attacks, with each attack consisting of an initial attack and follow-up attacks over $t_{\mathrm{WP}} = 3$ timesteps. The total number of attack steps $K$ for Wolfpack attack is then calculated as $K = 4 \times K_{\mathrm{WP}}$. In this section, we conduct a parameter search for $K_{\mathrm{WP}} \in [1, 2, 3, 4]$. Fig. F.1 illustrates the robustness of WALL policies trained with different $K_{\mathrm{WP}}$ values under the default Wolfpack attack in 8m and MMM. In both environments, having too small $K_{\mathrm{WP}}$ results in insufficiently severe attacks, which leads to reduced robustness of the WALL framework. Conversely, in the 8m environment, excessively large $K_{\mathrm{WP}}$ values create overly devastating attacks, making it difficult for CTDE methods to learn strategies to counter the Wolfpack attack, which degrades learning performance. Therefore, an optimal $K_{\mathrm{WP}}$ exists in both environments: $K_{\mathrm{WP}} = 2$ for 8m and $K_{\mathrm{WP}} = 4$ for MMM, which we choose as the default hyperparameters.

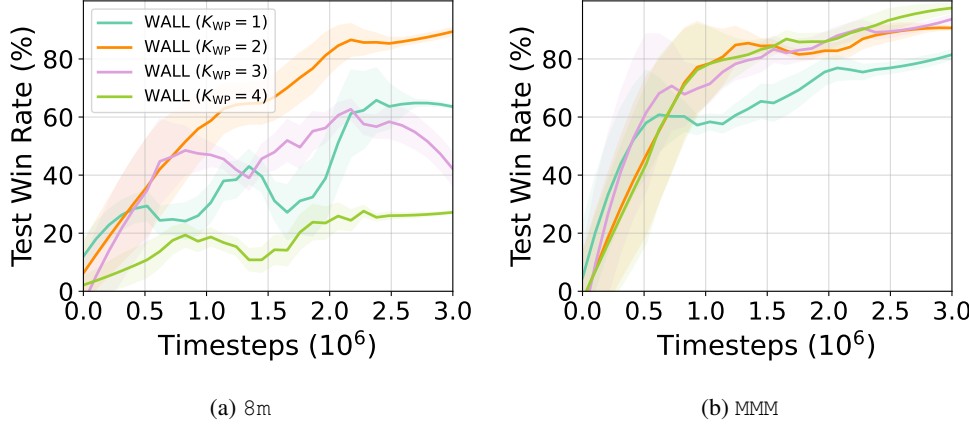

(a) 8m

(b) MMM

*Figure F.1.* Number of Wolfpack attacks ($K_{\mathrm{WP}}$)

**Attack duration** $t_{\mathrm{WP}}$**:** The hyperparameter $t_{\mathrm{WP}}$ determines the duration of follow-up attacks after the initial attack. To analyze its impact on robustness, Fig. F.2 compares performance for $t_{\mathrm{WP}} \in [1, 2, 3, 4]$ in the 8m and MMM environments. As shown in the figure, similar to the case of $K_{\mathrm{WP}}$, setting $t_{\mathrm{WP}}$ too low results in insufficient follow-up attacks on assisting agents, reducing the severity of the attack and lowering the robustness of WALL. On the other hand, excessively high $t_{\mathrm{WP}}$ values lead to overly severe attacks, making it challenging for WALL to learn effective defenses against the Wolfpack attack. Both environments demonstrate that $t_{\mathrm{WP}} = 3$ yields optimal performance and is selected as the best hyperparameter.

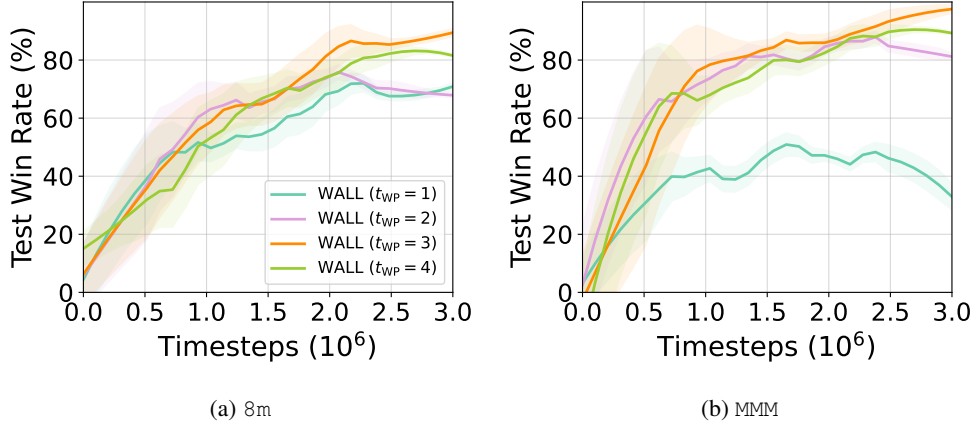

(a) 8m

(b) MMM

*Figure F.2.* Attack duration ($t_{\mathrm{WP}}$)

# G. Additional Visualizations of Wolfpack Adversarial Attack

## G.1. Visualization of Wolfpack Adversarial Attack Across Additional SMAC Scenarios

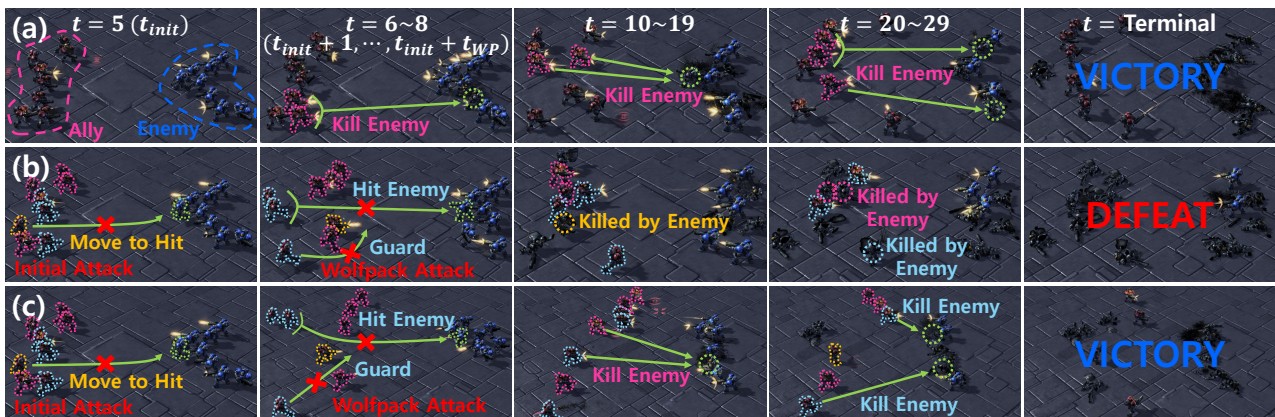

*Figure G.1.* Attack comparison on 8m task in the SMAC: (a) QMIX/Natural, (b) QMIX/Wolfpack attack, and (c) WALL/Wolfpack attack

To analyze the superior performance of the Wolfpack attack, we provide a visualization of its execution in various SMAC environments. Fig. 8 illustrates the 2s3z task, while Fig. G.1 visualizes the 8m task, and Fig. G.2 presents the MMM task.

Fig. G.1(a) illustrates Vanilla QMIX operating in a natural scenario without any attack, where the agents successfully defeat all enemy units and achieve victory. In this scenario, agents with low health continuously move to the backline to avoid enemy attacks, while agents with higher health position themselves at the frontline to absorb damage. This dynamic coordination enables the team to manage their resources effectively, withstand enemy attacks, and secure a successful outcome.

Fig. G.1(b) depicts Vanilla QMIX under the Wolfpack adversarial attack, where an initial attack is launched at $t = 5$, and follow-up agents are targeted between $t = 6$ and $t = 8$. Agents with higher remaining health are selected as follow-up agents, preventing them from guarding the targeted ally or engaging the enemy effectively. Between $t = 10$ and $t = 19$, the initial agent continues to take focused enemy fire, eventually succumbing to the attacks and being eliminated. This disruption renders the remaining agents ineffective in defending against the adversarial attack, leading to a loss as all agents are defeated.

Fig. G.1(c) shows the policy trained with the WALL framework. During $t = 6$ to $t = 8$, the same agents as in (b) are selected as follow-up agents and subjected to the Wolfpack attack, limiting their ability to guard or engage the enemy. Nevertheless, the non-attacked agents adjust by forming a wider formation vertically, effectively dispersing enemy firepower while delivering coordinated attacks. Additionally, agents with higher health move forward to guard the initial agents, ensuring that the initial agents do not die. This tactical adaptation enables the team to eliminate enemy units and secure victory.

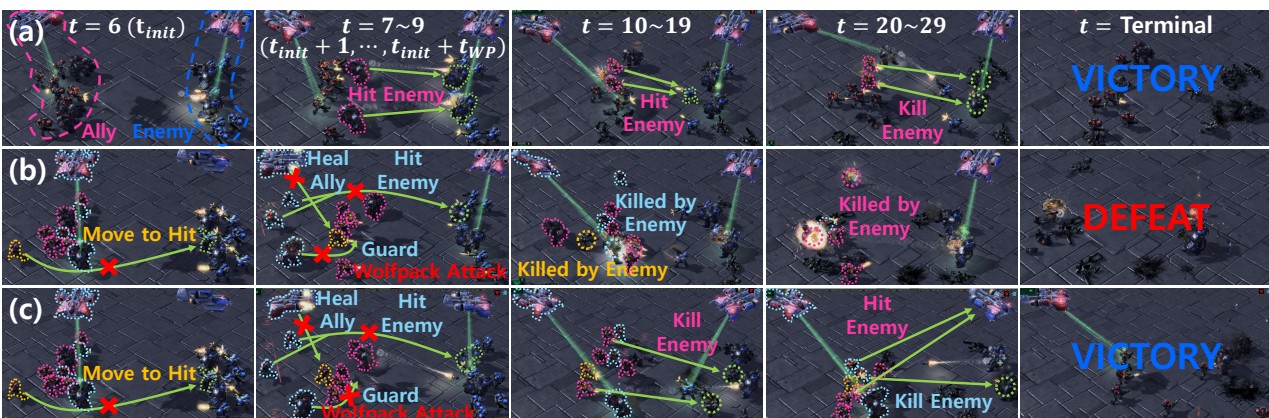

*Figure G.2.* Attack comparison on MMM task in the SMAC: (a) QMIX/Natural, (b) QMIX/Wolfpack attack, and (c) WALL/Wolfpack attack

Fig. G.2(a) showcases Vanilla QMIX operating in a natural scenario without any adversarial interference, where all enemy units are successfully eliminated, leading to a decisive victory. During this process, agents with lower health retreat to the back while the Medivac agent provides healing, and agents with higher health move forward to absorb enemy attacks. This coordinated strategy enables the team to secure victory efficiently.

Fig. G.2(b) illustrates Vanilla QMIX under the Wolfpack adversarial attack. An initial attack is launched at $t = 6$, followed by the targeting of four follow-up agents between $t = 7$ and $t = 9$. The follow-up agents selected include one healing agent attempting to heal the initial agent, one guarding agent positioned to protect the initial agent, and two agents actively engaging the enemy targeting the initial agent. Due to the Wolfpack attack, the initial agent failed to receive critical healing or guarding support at $t = 10$ and $t = 19$, leading to its elimination. This disruption severely hinders the remaining agents' ability to defend against the adversarial attack, ultimately resulting in a loss as all agents are defeated.

Fig. G.2(c) illustrates the policy trained with the WALL framework. During $t = 7$ to $t = 9$, the same agents as in (b) are selected as follow-up agents and subjected to the Wolfpack attack, restricting their actions such as healing, guarding, or targeting the enemy effectively. However, the non-attacked agents adapt by positioning themselves ahead of the initial agent to provide protection and focus their fire on the enemies targeting the initial agent. This strategic adaptation enables the team to successfully repel the adversarial attack, eliminate enemy units, and secure victory. Notably, in the terminal timesteps, more agents survive under the WALL framework compared to the natural scenario depicted in (a), highlighting the enhanced robustness and stability of the policy learned with WALL.

## G.2. Additional Analysis of Follow-up Agent Group Selection

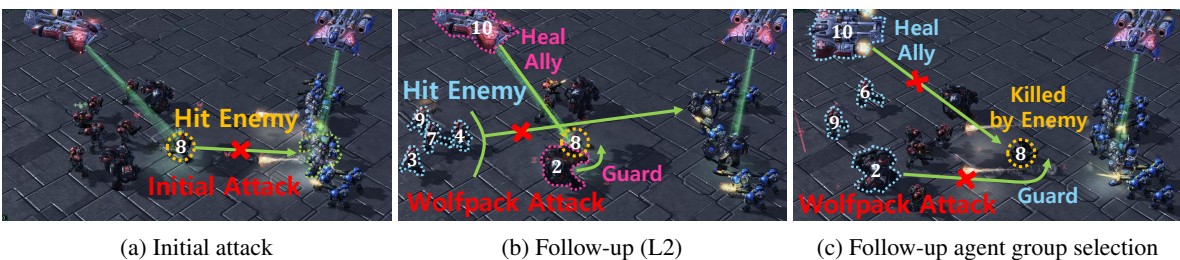

(a) Initial attack          (b) Follow-up (L2)         (c) Follow-up agent group selection

*Figure G.3.* Visualization of follow-up agent group selection comparison for the MMM task in SMAC

In this section, we demonstrate that the proposed Follow-up Agent Group Selection method effectively identifies responding agents that protect the initially attacked agent. Fig. G.3 visualizes the process of selecting the follow-up agent group after an initial attack. By comparing the proposed method with a baseline method, Follow-up (L2), which selects $m$ agents closest to the initial agent based on observation L2 distance, we show that our method better identifies responding agents, enabling a more impactful Wolfpack attack.

Fig. G.3(a) illustrates an initial attack on agent $8$, preventing it from performing its original action of hitting the enemy and forcing it to move forward, exposing it to enemy attacks. Fig. G.3(b) shows the Follow-up (L2) method selecting agents $3, 4, 7, 9$ as the follow-up agents based on their proximity to the initial agent. Despite the follow-up attack, non-attacked agents, which is far from the initial agent in terms of observation L2 distance, such as agent $10$, heal the initial agent, while agent $2$ guards it, effectively protecting the initial agent from the attack.

Fig. G.3(c) illustrates the follow-up agents selected using our proposed Follow-up Agent Group Selection method. The selected group includes agents $2$ and $10$, which are responsible for healing and guarding the initial agent, and agents $6$ and $9$, which are hitting enemies targeting the initial agent. These agents are subjected to the follow-up attack, preventing them from performing their protective actions. As a result, the initial agent is left vulnerable, succumbs to enemy attacks, and is ultimately eliminated.

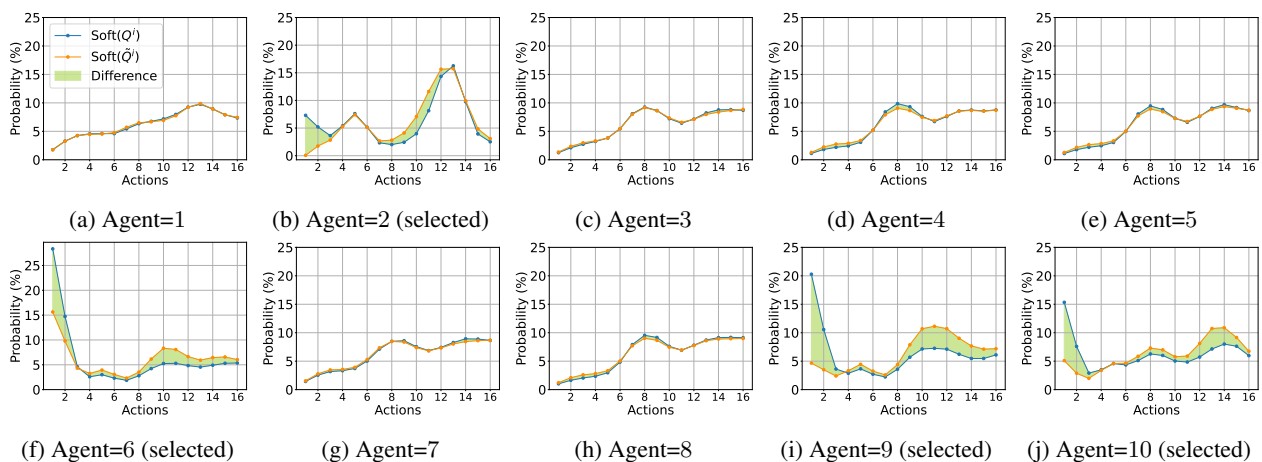

*Figure G.4.* $Soft(Q^i)$ and $Soft(\tilde{Q}^i)$ for each agent, along with the difference between the two distributions.

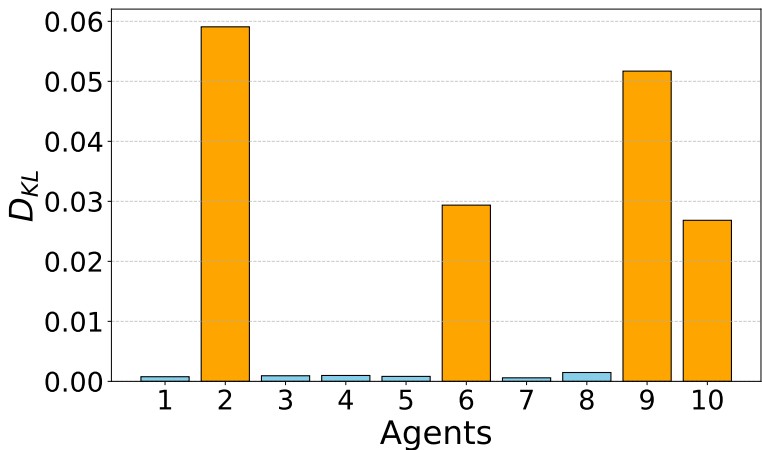

*Figure G.5.* KL divergence values for each agent, representing the difference between $Soft(Q^i)$ and $Soft(\tilde{Q}^i)$ distributions.

Fig. G.4 illustrates the $Soft(Q^i)$ distribution and the updated $Soft(\tilde{Q}^i)$ distribution based on Equation 1, highlighting the differences between the two distributions. It is evident that agents $2, 6, 9, 10$ exhibit the largest differences in their distributions. This suggests that, following the initial attack, these agents show noticeable policy changes to adapt and defend against it. Additionally, Fig. G.5 presents the KL divergence values between these two distributions, further confirming that agents $2, 6, 9, 10$ have the highest KL divergence values.

Consequently, based on Equation 2, agents $2, 6, 9, 10$ are selected as the follow-up agent group. This aligns with the visualization results shown in Fig. G.3, demonstrating consistency in the SMAC environment.

