# OpenReview forum: "Wolfpack Adversarial Attack for Robust Multi-Agent Reinforcement Learning"
_ICML.cc/2025/Conference — ICML 2025 poster_

### Official Review · Reviewer_z86J · 2025-02-18

**Overall Recommendation:** 2

**Summary:**

This paper propose Wolfpack adversarial attack for robust MARL, which targets not only the initial perturbed agent, but also the agents that follow-up to help this perturbed agent. Based on wolfpack attack, the author further propose the wolfpack-adversarial learning for MARL, which learns robust policy based on wolfpack attack. Experiment results show the defense is more robust, and the attack is more harmful.

**Claims And Evidence:**

Yes.

**Essential References Not Discussed:**

Most important references are discussed.

**Experimental Designs Or Analyses:**

Yes, experiments are comprehensive and solid. However, I do have concerns on the perturbation budget used.

**Methods And Evaluation Criteria:**

Yes, albeit in a narrow scope.

**Other Comments Or Suggestions:**

Please see "Other Strengths And Weaknesses"

**Other Strengths And Weaknesses:**

Strength.

1. I would appreciate the strong empirical focus of this paper, which identifies the current weakness of many robust MARL algorithms. While one agent is being attacked, other agents trained by robust MARL would come to help it. Wolfpack attack responds to this by also targeting these agents as well, which is an interesting insight provided in this paper.

2. I appreciate the extensive empirical evaluation in the paper, which indicates the effectiveness of the method proposed in this paper.

3. The paper is clearly written and easy to follow.

Weakness.

1. The biggest weakness I believe is the scope of this paper. While robust MARL is already a narrow field, the threat model proposed by LPA-Dec-POMDP is one of the many threats in robust MARL. The definition of Wolfpack attack, while focusing on action uncertainties, is another narrower field: it first assumes the LPA-Dec-POMDP threat model, then base its attack on it. What if other form of threat model and formulations is applied? How can Wolfpack attack based on LPA-Dec-POMDP formulation generalize to other settings?

2. While I do not deny wolfpack attack is interesting, the entire paper is more like a follow-up work of Yuan et al (2023) and its journal version, ie, using same framework and just an updated attack which requires more perturbation budget. Since the attacker is using larger perturbation budget, it is natural that existing defenses are broken by the attack and the proposed attack is the most robust. What about these robust methods trained with larger perturbation budgets? or any discussions?

3. While I appreciate the strong empirical performance of the paper, I do believe there is a lack of theoretical analysis, including convergence, equilibrium, etc, which better fits the scope of ICML.

4. This is minor. The definition applies only to QMIX-family, the application to MAPPO family is not straightforward, but can be imagined. Maybe adding a discussion in conclusion or section 3.1 helps.

**Questions For Authors:**

Please see "Other Strengths And Weaknesses"

**Relation To Broader Scientific Literature:**

The paper is a part of robust MARL literature, but more specifically related to the LPA-Dec-POMDP paper in a reference.

**Theoretical Claims:**

The author do not make theoretical claims.

---

> ### Author Rebuttal · Authors · 2025-03-31
>
> We sincerely thank the reviewer for their thoughtful and constructive comments. We greatly appreciate the opportunity to address the main concerns regarding the scope of our method, the perturbation budget, and the theoretical contributions. We provide detailed responses to each of these points below.
>
> **Scope of our method:**
>
> Our proposed Wolfpack adversarial attack introduces a novel class of adversarial attacks in MARL that we believe goes beyond existing approaches. Unlike ROMANCE, which focuses on building a population of diverse attackers targeting a single agent (EGA), Wolfpack explicitly models **multi-step attacks targeting multi-agents**. Specifically, it targets not only a single agent but also its cooperating teammates in follow-up time steps, thereby exposing relational vulnerabilities in agent coordination that are not captured by single-agent or uncoordinated perturbations.
>
> A key innovation also lies in our use of a **Transformer-based attack planner** for critical step selection. This module allows the attacker to identify time steps where coordinated interventions lead to maximal long-term disruption, particularly important when attacks are spread over multiple steps. ROMANCE, by contrast, does not perform planning-based or temporally extended attacks. We hope this distinction clarifies that our method is not a straightforward extension of ROMANCE but rather proposes a new class of coordinated adversarial behavior in MARL.
>
> Although we adopt the LPA-Dec-POMDP formulation in our training setup, we emphasize that this choice is primarily for structuring the learning process with a finite attack budget and enabling convergence guarantees. Importantly, our method is not inherently restricted to the LPA-Dec-POMDP setting. Since our planner enables the policy to anticipate attacks based on learned dynamics, WALL remains effective even without explicitly encoding the attack budget, suggesting general applicability beyond this specific framework. We hope this addresses the reviewer’s concern regarding the generality of our approach.
>
> **Theoretical contributions:**
>
> We understand and appreciate the reviewer’s concern regarding the lack of formal convergence or equilibrium analysis. As discussed above, our work emphasizes practical effectiveness in attacker modeling and defense under realistic MARL coordination settings. In particular, we focus on demonstrating how temporally extended, learning-based attackers can expose structural weaknesses in existing robust MARL methods.
>
> While some works in robust MARL pursue theoretical guarantees, a substantial body of literature (e.g., [1–2]) actively explores attacker design and empirical robustness under realistic conditions. Our approach aligns with this latter direction: the use of Transformer-based planning makes formal convergence analysis challenging, but reflects the state of practice in robust MARL where empirical insights are often more actionable than theoretical ones. However, we agree that developing theoretical underpinnings for planning-based attacks is a valuable and non-trivial direction. We consider this a promising line of future research.
>
> [1] Gleave et al., "Adversarial policies: Attacking deep reinforcement learning," ICLR 2020.
>
> [2] Ao et al., "Eat-c: environment-adversarial sub-task curriculum for efficient reinforcement learning," ICML 2022.
>
> **Perturbation budget comparisons:**
>
> We appreciate the reviewer’s concern about the role of perturbation budget. However, as our experiments show, attack budget does not directly correlate with robustness. For example, in both ROMANCE (Yuan et al.) and our own method, increasing the number of attacks can degrade policy learning, as overly perturbed environments prevent effective strategy formation.
>
> In our case, as shown in Fig. 9 and Fig. F.1, robustness does not improve monotonically with more attack steps or follow-up agents. Instead, we perform parameter sweeps for each method and compare their performance under optimal attack configurations. Thus, our comparisons are fair and based on each method’s best-case performance. The success of Wolfpack arises not from larger attack volume, but from coordinated timing and targeting, which prior defenses fail to counter.
>
> **Applicability to MAPPO:**
>
> Although our work focuses on value-based CTDE methods, the Wolfpack framework can be extended to actor-critic approaches like MAPPO, which also follow the CTDE paradigm. While some architectural adjustments (e.g., adapting the $Q$-value estimation) would be needed, the core ideas of coordinated attack modeling and robust training remain applicable. We will note this in the revised version as a potential direction for future work.
>
> Once again, we sincerely thank the reviewer for their careful reading and insightful feedback. We hope that our clarifications and responses adequately address the raised concerns and help situate our contributions within the broader scope of robust MARL research.

---

### Official Review · Reviewer_YyTB · 2025-03-12

**Overall Recommendation:** 3

**Summary:**

This paper proposes a Wolfpack adversarial attack to train robust MARL systems. Specifically, it modifies the attack process to attack an agent and its follow-up teammates. Experiments on SMAC benchmarks demonstrate the effectiveness of the proposed method.

**Claims And Evidence:**

Yes

**Essential References Not Discussed:**

The purterbation of observations in MARL is an open-MARL problem, where the key factors of the learning process change. A survey of this problem [1] should be discussed.

[1] A Survey of Progress on Cooperative Multi-agent Reinforcement Learning in Open Environment. Lei Yuan, et al. 2023.

**Experimental Designs Or Analyses:**

Yes. I checked Section 5. Experiments.

**Methods And Evaluation Criteria:**

Yes

**Other Comments Or Suggestions:**

None.

**Other Strengths And Weaknesses:**

## Strengths
1. **Clarity and Accessibility**: The paper is well-written and easy to follow. The proposed method, WALL, is presented in a clear and straightforward manner, making it accessible to readers.
2. **Novelty**: The idea of transforming single-step, fixed-agent attacks into consecutive attacks on follow-up agents is quite novel.
3. **Comprehensive Experiments**: The experiments are thorough and span diverse multi-agent attack scenarios and robust baselines. The results demonstrate consistent improvements over strong baselines, underscoring the effectiveness of the proposed method.

## Weaknesses
1. **Computational Cost**: WALL introduces a transformer-based module to select follow-up agents, which likely increases the computational overhead. However, the paper does not discuss or quantify this additional cost, leaving a gap in the practical evaluation of the method.
2. **Limited Environments**: While the experiments are robust, they are only conducted in the SMAC environment. Including additional environments or benchmarks would strengthen the generalizability and significance of the results.
3. **Presentation Issues**:
   - The abstract could be more informative, providing additional details about the key contributions and results.
   - Equation 1 appears to have an extra $\Delta$ symbol, which should be corrected.
   - In Line 186, the notation $\tilde{a}_t$ is incorrectly written as $\tilde{a}t$.

**Questions For Authors:**

See Weaknesses.

**Relation To Broader Scientific Literature:**

The key contributions extend and refine ideas from MARL, robustness in machine learning, and open-environment learning. By addressing the challenge of robustness in MARL, this paper advances these fields and provides a framework for developing more robustly collaborative AI systems.

**Theoretical Claims:**

There is no formulated theoretical claims in this paper.

---

> ### Author Rebuttal · Authors · 2025-03-31
>
> We sincerely thank the reviewer for their thoughtful and constructive comments. We greatly appreciate the opportunity to address the main concerns regarding computational cost and environmental diversity, and we provide detailed responses below.
>
> **Extra computational cost:**
>
> Thank you for pointing this out. To address the reviewer’s concern, we measured the total training time (for 3M steps) required for each algorithm in the SMAC environments. The results show that vanilla QMIX takes 6h 30m, ROMANCE requires 16h 36m, and WALL takes 21h 05m on average. Compared to QMIX, ROMANCE and WALL incur a 2.46× and 3.23× increase in training time, respectively. Also, WALL requires approximately 30% more time than ROMANCE, primarily due to the use of the Transformer. However, despite this additional cost, the Transformer-based critical step selection plays an essential role in the effectiveness of our method, as it enables precise identification of attack timing where coordinated perturbations are most disruptive. This capability allows the robust policy to anticipate and defend against strategically timed adversarial threats, ultimately resulting in stronger robustness.
>
> As shown in our component ablation study (Fig. 8), removing this module and replacing it with random step selection results in a substantial drop in robustness across all SMAC scenarios. This clearly demonstrates that the Transformer-based planner is not merely an architectural enhancement, but a core component contributing to the success of WALL. Furthermore, as illustrated in the training curves in Fig. 6, other methods do not reach WALL’s level of performance even with extended training time, indicating that the proposed method offers superior robustness despite its slightly higher computational cost. We hope this helps clarify and address the reviewer’s concern.
>
> **Environment diversity:** (Common response with Reviewer rqE9)
>
> We also appreciate the reviewer’s valuable suggestion to expand beyond SMAC in our evaluations. Our initial choice of SMAC was motivated by its widespread use as a benchmark in MARL, its cooperative partial observability, and its complex unit-level coordination dynamics. These characteristics make SMAC a rigorous and well-established testbed for adversarial robustness in decentralized learning.
>
> To further address the reviewer’s concern and support the broader applicability of our method, we conducted additional experiments in the Multi-Agent Particle Environment (MPE) [1], a widely adopted benchmark in MARL, focusing on three cooperative scenarios: tag 3/1, tag 6/2, and tag 9/3. These environments require agents to coordinate spatially to pursue adversarial targets, which closely aligns with the core motivations of our work. For comparison, attack-based methods were evaluated using their respective optimal hyperparameter setups, ensuring a fair and controlled evaluation across environments.
>
> As summarized in Table R.2, WALL consistently outperforms prior methods in these diverse MPE tasks. These results are included to address the reviewer’s concerns regarding environment diversity and to clarify that the effectiveness of our approach is not limited to the SMAC benchmark.
>
> **Table R.2:** Performance comparison on MPE envrionments.
>
> |Attacks|Methods|tag 3/1|tag 6/2|tag 9/3|Mean|
> |---|---|---|---|---|---|
> |Random Attack|Vanila QMIX|158.9|522.6|656.4|445.9|
> ||RANDOM|170.1|638.7|657.9|488.9|
> ||ROMANCE|173.4|624.2|701.7|499.7|
> ||WALL|**202.9**|**654.0**|**802.5**|**553.1**|
> |EGA|Vanila QMIX|153.7|499.8|585.1|412.8|
> ||RANDOM|157.6|604.6|589.7|450.6|
> ||ROMANCE|167.1|605.1|594.4|455.5|
> ||WALL|**166.5**|**685.5**|**682.7**|**511.5**|
> |Wolfpack attack|Vanila QMIX|135.7|415.6|551.7|367.6|
> ||RANDOM|157.5|571.0|624.3|450.9|
> ||ROMANCE|159.3|554.6|570.3|428.0|
> ||WALL|**171.5**|**599.0**|**698.1**|**489.5**|
>
> Additionally, in response to Reviewer rqE9’s suggestion regarding broader definitions of robustness, we also evaluated the robustness of WALL under observation noise and parameter noise conditions. For completeness, we kindly refer the reviewer to our response to Reviewer rqE9 for the full details of these experiments. We include these results to directly address the reviewer’s concern and will incorporate both the findings and discussion in the revised version.
>
> [1] Lowe et al., "Multi-agent actor-critic for mixed cooperative-competitive environments," NeurIPS 2017.
>
> **Presentation issues**
>
> We thank the reviewer for pointing out the issues related to clarity and formatting. We will revise the abstract to more clearly highlight our key contributions and results, and we will correct the formatting errors in Equation 1 and Line 186.
>
> Once again, we sincerely thank the reviewer for their careful reading, insightful comments, and positive recommendation. We hope that the additional experiments and clarifications adequately address the raised concerns and further reinforce the value of our work.

---

### Official Review · Reviewer_rqE9 · 2025-03-12

**Overall Recommendation:** 5

**Summary:**

This paper introduces a new kind of adversarial attack, Wolfpack, and trains policies against those attacks. They find that trained models are not only more robust against Wolfpack attacks but are also more robust to other adversarial attacks (EGA) and noise injection (Random Attack).

Wolfpack builds upon previous work (EGA) that attacks a single agent from the MARL set-up: after the first agent is attacked (has its action replaced), “exposed” agents are identified and are also attacked. In most settings, previous robustness methods (that are not designed to be robust to Wolfpack attacks) fail to be robust to the attack.

The paper provides empirical evidence with comprehensive evidence from the Starcraft II Multi-Agent Challenge environment.

**Claims And Evidence:**

The paper claims that WALL “significantly improves robustness compared to existing methods”. It provides evidence that WALL is more robust to action-replacement-based adversarial attacks (Wolfpack, EGA, Random Attack) than two prior methods for action-replacement adversarial attacks (ERNIE, ROMANCE) and non-action-replacement methods (e.g,. RARL and RAP). If these other methods are designed to deal with other kinds of adversaries, evidence that other methods are not robust to Wolfpack attacks may not, in itself, substantiate that WALL improves “robustness” in a broader sense, based on evidence from Wolfpack attacks.

However, the authors also show that WALL is more robust to random noise in action replacement than other methods, which I believe is significantly more compelling evidence that the method is more robust in general. Overall, I find the range of alternative methods, environments, and attack types to provide convincing evidence that WALL is more robust.

The authors claim that their framework is novel, and they support this claim with a comprehensive review of the literature.

**Essential References Not Discussed:**

No known missing references.

**Experimental Designs Or Analyses:**

I did not find any problems in the soundness or validity of the experimental design.

**Methods And Evaluation Criteria:**

One main stated motivation of the work is “enhancing the robustness of CTDE”, especially to address “challenges from mismatches between training and deployment environments”. However, the evaluation of the new method is largely against adversarial attacks, particularly those where actions are replaced. The authors do measure robustness against random noise (Random Attack), but this is only against the random replacement of attacks and does not consider other threat models like noisy observations or distributional shift in the underlying Dec-POMDP between training and deployment.
While this may be a common move in the literature, if the primary contribution is improved robustness to adversarial attacks, I’d like to see either an argument or evidence that this helps with generalization to deployment. Alternatively, the authors could justify why their attack model is relevant.

The choice of the SMAC environment and the choice of benchmark training methods is appropriate. However, I think a more convincing demonstration would use additional environments (beyond SMAC)  to increase diversity.

Overall, I find that the paper’s proposed evaluation criteria are for justifying robustness when restricted to some adversarial attacks and random noise.

**Other Comments Or Suggestions:**

There are some typos, for example: “Notably, despite RANDOM is trained against Random Attack and ROMANCE against EGA, WALL achieves better performance against both attack types.”

**Other Strengths And Weaknesses:**

Strengths:
* A major strength of this paper’s contributions is the improvement of WALL above state-of-the-art methods on Random Attack. Given the differences between Random Attack and the Wolfpack adversarial attacks, the generalisation to improve performance is impressive.
* The paper thoroughly investigates the robustness of WALL, with substantial evidence provided in a relevant and challenging environment.
* The breadth of environment, attack type, and robust training methods is particularly compelling.
* The paper was presented extremely clearly, with a thorough analysis of previous literature and a clear and full explanation of the relevant background.

Weaknesses:
* As stated above, I think that the paper could have shown its contributions were even more significant for improving robustness to train-test distribution shift. For example, they could compare their method to others in a SMAC environment with different parameterization during test-time than during train-time.

**Questions For Authors:**

None

**Relation To Broader Scientific Literature:**

The relation to broader scientific literature is thorough, with a wide range of previous work being explained and methods compared across benchmarks.

**Theoretical Claims:**

No theoretical claims are provided.

---

> ### Author Rebuttal · Authors · 2025-03-31
>
> We sincerely thank the reviewer for their thoughtful and constructive comments. We greatly appreciate the opportunity to address the main concerns regarding general robustness and environmental diversity, and we provide detailed responses below.
>
> **General robustness beyond action perturbations:**
>
> We appreciate the reviewer’s insightful suggestion to consider broader notions of robustness beyond action-level perturbations. While our work primarily focuses on robustness against coordinated adversarial attacks and random action replacement, we fully agree that a comprehensive understanding of robustness should also account for other forms of distributional shift, such as observation noise and parameterization mismatches in Dec-POMDPs, which are critical for real-world deployment.
>
> Our choice to focus on action-level adversarial attacks is based on their prevalence and practical relevance in the MARL literature. Prior works in robust MARL (e.g., [1–2]) commonly adopt this setting, as action perturbations directly affect decentralized decision-making, where each agent selects actions based solely on local observations and individually learned policies. Building upon this foundation, our contribution introduces coordinated multi-agent action perturbations, an attack model designed to reveal cooperation-disrupting vulnerabilities not exposed by existing single-agent or uncoordinated attacks.
>
> To address the reviewer’s concern, we conducted additional experiments to evaluate robustness under other commonly used criteria in the 8m and MMM environments. Specifically, we considered: **(1) Gaussian observation noise:** Standard Gaussian noise (mean 0, standard deviation 1) is injected into agents’ local observations, where attack steps (total attack steps $K$: 8 for 8m, 16 for MMM) and the attack group are selected randomly; and **(2) Different parameterization in the SMAC test environment:** Robustness is evaluated under perturbed unit attributes by reducing the initial health of allied units by 10%, 15%, and 20% in the test setup, compared to the training configuration. These types of noise and perturbations are standard in robustness evaluations and enable us to assess how well the proposed method generalizes beyond action-level perturbations.
>
> The Table R.1 below summarizes the results. In both settings, WALL demonstrates consistently stronger performance than existing baselines, suggesting that it generalizes well to broader forms of distributional shift.
>
> **Table R.1:** Performance comparison under various perturbation setups in SMAC environments.
>
> | Setting | Methods | 8m  | MMM  | Mean |
> | --- | --- | --- | --- | --- |
> | Gaussian Obs. Noise | Vanila QMIX | 62.6 | 75.3 | 69.0  |
> |  | RANDOM | 72.3 | 79.3 | 75.8  |
> |  | ROMANCE | 69.6 | 76.6 | 73.1  |
> |  | WALL (ours) | **91.3** | **97.3** | **94.3**  |
> | Ally HP ↓ 10% | Vanila QMIX | 52.8 | 88.4 | 70.6  |
> |  | RANDOM | 50.8 | 95.4 | 73.1  |
> |  | ROMANCE | 56.8 | 92.4 | 74.6  |
> |  | WALL (ours) | **73.2** | **98.6** | **85.9**  |
> | Ally HP ↓ 15% | Vanila QMIX | 47.4 | 69.0 | 58.2  |
> |  | RANDOM | 49.2 | 81.4 | 65.3  |
> |  | ROMANCE | 57.6 | 81.2 | 69.4  |
> |  | WALL (ours) | **69.2** | **94.0** | **81.6**  |
> | Ally HP ↓ 20% | Vanila QMIX | 0.3 | 41.2 | 20.8  |
> |  | RANDOM | 0.0 | 65.0 | 32.5  |
> |  | ROMANCE | 2.3 | 57.0 | 29.7  |
> |  | WALL (ours) | **4.3** | **89.6** | **47.0**  |
>
> [1] Yuan et al., "Robust multi-agent coordination via evolutionary generation of auxiliary adversarial attackers," AAAI 2023.
>
> [2] Li et al., "Byzantine robust cooperative multi-agent reinforcement learning as a bayesian game," ICLR 2024.
>
> **Environment diversity:** (Common response with Reviewer YyTB)
>
> Thank you for raising the important point regarding environment diversity. In line with ICML’s policy of one rebuttal per reviewer and encouraging shared responses where applicable, we kindly refer you to our detailed response under Reviewer YyTB, where we provide additional performance comparisons on the cooperative Multi-Agent Particle Environment (MPE) [3], a widely adopted benchmark in MARL. We appreciate your understanding.
>
> [3] Lowe et al., "Multi-agent actor-critic for mixed cooperative-competitive environments," NeurIPS 2017.
>
> **Minor issues:**
>
> We thank the reviewer for pointing out the typo and will correct it.
>
> Once again, we sincerely thank the reviewer for their careful reading, insightful comments, and positive recommendation. We hope that the additional experiments and clarifications meaningfully address the reviewer’s concerns and further strengthen the impact of our work.

---

### Decision · Program_Chairs · 2025-05-01

**Decision:**

Accept (poster)

**Comment:**

In this submission, the authors propose the Wolfpack Adversarial Attack framework, which targets an initial agent and its assisting agents to disrupt cooperation in multi-agent settings.  They also proposed the Wolfpack-Adversarial Learning for MARL (WALL) approach that can be used to train MARL policies to defend against these types of attacks. The authors provide empirical results which showcase the effectiveness of the proposed methods, both in terms of attacking as well as defending against it.

All of the reviewers believe that this work demonstrates strong empirical performance. They also highlight the novelty of the approach and the comprehensive set of experiments included in the manuscript. The reviewers also noted various limitations of the original paper, including the inclusion of only a single environment, the lack of discussion on computational costs, and the robustness of the approach (beyond action perturbations). The authors have addressed many of the reviewers' concerns during the rebuttal, which has led to an increase in scores.

Overall Assessment: Based on the reviewers' comments and discussions during the rebuttal, I believe this paper makes a strong technical contribution. I hope the reviewers’ comments and the results shown during the rebuttal can help the authors prepare an improved version of this submission for camera-ready.